# A Critical Review of the Current Global Ex Situ Conservation System for Plant Agrobiodiversity. II. Strengths and Weaknesses of the Current System and Recommendations for Its Improvement

**DOI:** 10.3390/plants10091904

**Published:** 2021-09-14

**Authors:** Johannes M. M. Engels, Andreas W. Ebert

**Affiliations:** 1Alliance of Bioversity International and CIAT, 00153 Rome, Italy; 2World Vegetable Center, 60 Yi-Min Liao, Tainan 74151, Taiwan; ebert.andreas6@gmail.com

**Keywords:** plant agrobiodiversity, routine gene bank operations, active collection, base collection, linking conservation and use, genomics, phenomics, conservation strategies, global conservation network

## Abstract

In this paper, we review gene bank operations that have an influence on the global conservation system, with the intention to identify critical aspects that should be improved for optimum performance. We describe the role of active and base collections and the importance of linking germplasm conservation and use, also in view of new developments in genomics and phenomics that facilitate more effective and efficient conservation and use of plant agrobiodiversity. Strengths, limitations, and opportunities of the existing global ex situ conservation system are discussed, and measures are proposed to achieve a rational, more effective, and efficient global system for germplasm conservation and sustainable use. The proposed measures include filling genetic and geographic gaps in current ex situ collections; determining unique accessions at the global level for long-term conservation in virtual base collections; intensifying existing international collaborations among gene banks and forging collaborations with the botanic gardens community; increasing investment in conservation research and user-oriented supportive research; improved accession-level description of the genetic diversity of crop collections; improvements of the legal and policy framework; and oversight of the proposed network of global base collections.

## 1. Introduction

Plant agrobiodiversity, i.e., the diversity of plants that is used or has the potential to be used in agriculture and horticulture, has been the foundation for human subsistence for millennia and will continue to play a decisive role in securing global food and nutrition security for a growing population, especially under the current threat of climate change. While 7039 edible plant species are known to science and 417 are considered food crops [1], today, only 12 plant and five animal species are used to supply 75% of human food [2]. This over-reliance of global food production on a very small number of crops and animals with a largely very limited number of genetically uniform, high-yielding varieties of crop plants and breeds of animals presents a major challenge for both the conservation of agrobiodiversity and for human nutrition and health.

While we (still) enjoy such an enormous agrobiodiversity, we are being made aware that two out of five plant species are threatened with extinction according to current estimates [1]. The major threats of genetic erosion of plant species are of anthropogenic nature and include agriculture and modern plant breeding; overexploitation of biological resources in the wild; modification, fragmentation, and destruction of natural ecosystems; rapidly expanding residential and commercial developments; pollution; and climate change. Conscious of the threat of modern agriculture and mankind to (plant) agrobiodiversity, plant introduction centers evolved since the early 20th century in several countries that later grew out into gene banks. These efforts were made to meet the growing demand of plant breeders for broad genetic diversity, essential for the development of well-adapted, high-yielding varieties with resistance to biotic threats and tolerance to abiotic stresses. The history and the major players in the development of global long-term conservation practices and the evolving global (ex situ) conservation system have been described in a previous paper [3].

While aiming at an accurate description of the “global conservation system” that gradually emerged under the auspices of FAO, it has become clear that the individual components of that system evolved somewhat “spontaneously” and that no precise goal of that system existed. Consequently, not all components are logically embedded in the system and suffered adjustments to the “in parallel” evolving political framework and changing realities. Some components “disappeared”, and others were announced but did not materialize. Thus, the authors felt it was necessary to aim at a “working definition” of the global conservation system as follows: “A long-term global plant agrobiodiversity conservation system of well-defined national and international ex situ seed, tissue and plant collections that is managed under agreed genebank quality management standards and in harmony with the prevailing political framework regarding access and benefit-sharing, and that aims at safe, effective, efficient and rational long-term conservation and facilitating use by making high-quality accession-level information available”.

In this paper, we focus on the major routine gene bank activities and assess several constraints that might affect long-term ex situ conservation activities. A specific aspect of the current long-term conservation and facilitation of use “system” is the concept of base and active collections, as this concept has been designed in the past to address and resolve the difficult issue of linking conservation and use.

Having looked at the different conservation approaches as well as at the major ex situ gene bank management activities, it will be indispensable to describe the major components of the current global (long-term) conservation system and to identify and describe its strengths and weaknesses.

Molecular techniques, genomics, and bioinformatics have developed since the current global conservation system emerged and have only been applied to a limited extent in managing germplasm in gene banks and facilitating their use. In addition, communication technologies, digitalization capacities, bioinformatics, and possibly other recent developments could well provide opportunities for strengthening the current system and possibly help to rationalize the long-term conservation and the facilitation of the use of conserved materials.

This paper will focus on: (i) the description of the main conservation approaches and activities; (ii) their strengths and weaknesses; (iii) the advancements made in molecular genetics, genomics, bioinformatics, and the considerably increased knowledge regarding genetic diversity aspects; (iv) considerations regarding other developments that have an impact on long-term conservation and offer opportunities for possible improvements of the ex situ long-term conservation system, including the policy framework. Furthermore, suggestions and recommendations will be formulated on how the current global long-term conservation system can be made more rational and effective to allow more efficient conservation of plant agrobiodiversity.

## 2. Brief Description and Critical Review of Key Routine Germplasm Conservation Activities

In this section, we will focus on the major routine germplasm conservation activities, typically being part of gene bank operations across the world (see Figure 1). We critically review these operations with the intention to identify weak and/or critical aspects that should be considered to ensure optimum performance and thus contribute to an effective and efficient global conservation and sustainable use system. It is understood that other aspects such as sustainable funding of gene bank operations are important if not essential prerequisites for effective and efficient conservation and for facilitating the use of conserved germplasm materials. Funding, in particular, is such a complex, diverse and circumstantial aspect that would require specialized knowledge and expertise to be treated comprehensively, and to do this topic justice would blow the remit and scope of this paper. For details on the risk of decreasing funding and staffing of conservation and breeding programs, see the recent paper of Coe et al. [4].

### 2.1. Exploration and Collecting

Exploration of plant genetic resources is the first step after the decision has been made to conduct a collecting mission. One critically important aspect to check before making this decision is to see if the gene bank has sufficient capacity, adequate expertise, and financial resources for timely and effective regeneration and sustainable long-term storage or that maintenance of the collected materials is secured [7]. Exploration can be defined as “the act of searching or traveling around a terrain for the purpose of discovery of resources or information” [8]. The process of gathering information starts already well before a trip/mission is undertaken, through literature searches regarding the “where” (i.e., the areas and places where a given species naturally occurs, or from where it has been reported and/or is cultivated), “what” kind of material (i.e., species and populations or varieties to be collected) can be found, etc.

The decision to implement a mission should be based on the following actions:Contacting scientists in the areas concerned, asking for “on the ground information” about aspects such as species distribution, genetic diversity, time of seed or material maturity, genetic erosion, and threat status;Whether or not there have been collecting missions conducted in the past, where the collected materials are being conserved (in situ and/or ex situ), and if the materials would be readily available;Which sampling strategy had been used to obtain a better idea if “re-collecting” of wanted genetic diversity would be justified, also from a long-term conservation perspective;Targeted gap analysis on geographical coverage of the collection as well as genetic diversity presence in the collection; andPossibly other specific considerations.

Such a well-informed decision also includes aspects on which geographic area(s) to concentrate on, if a more refined search for specific information is needed through literature searches, local contacts, etc. This refined search should include the collecting of all available information on the population structure of the target species prior to the collecting mission [9,10].

The next step will be the search for collaborators and possibly specialists on the target species and, whenever possible, local plant collectors that would be prepared to join the exploration mission, to define the best period in the year to conduct the exploration/collection, to define what kind of outfit and equipment would be needed, etc. Finally, a formal request will have to be made to the local and national authorities to be allowed to explore (and possibly collect) plant genetic resources and thus, to be able to conclude a formal agreement for traveling to the identified sites and eventually to be allowed to take collected resources out of the country, when applicable (for details, see the work of [11]).

Frequently, exploration is combined with the actual collecting of the genetic resources taxa that had been prioritized by the conservation program to be added to the collection, for whatever reasons. Engels [12] distinguishes different reasons to collect as well as different types of collecting missions, i.e., (a) rescue collecting; (b) collecting for immediate use; (c) gap-filling for future use; (d) research; and (e) opportunistic collecting. Each of these types of collecting missions has different objectives, might require different sampling strategies, and might not all lead to an adequate representation of the genetic diversity for the target taxa in the collected samples. Furthermore, he also distinguishes several types of collecting missions, including multi-species vs. species-specific collecting, wild species vs. crop collecting, etc. These types require different preparations, different sampling strategies, etc. Another important aspect to ascertain before embarking on a collecting mission is to establish what the possible precise collecting sites are, i.e., natural habitats, disturbed areas, farming fields, marketplaces, home gardens, etc. in order to decide on the required (transport) equipment, accommodation, possible need to prepare meals, etc. and to determine what to collect where [12].

When in the field, taking careful notes on the collecting site (for a definition, see the work of [12]); deciding on the most effective sampling strategy (that might well vary with: a. the biology of the target taxon or taxa, e.g., annual/perennial, self- or outbreeder, seeds or vegetative parts to be collected; b. the precise purpose of collecting (see above); c. here we focus on collecting the maximum amount of genetically useful variability in the target species while keeping the number of samples within the practical limits for long-term conservation. Possibly the most important criterium is the frequency of alleles in the population, i.e., the conceptual class of the alleles: (1) common, geographically widespread, (2) rare, widespread, (3) common, localized, and (4) rare, localized. For a comprehensive treatment on this subject, please refer to the work of [9,10]. Besides genetic considerations, the number of individuals per population or variety to be collected also depends on the “viability” of the materials collected and on how many gene banks or collections are expecting subsamples of the collected materials (each with the same genetic diversity). Materials collected in the field need to be treated with great care to avoid that the viability of the collected organs (seeds, cuttings, tubers, etc.) would drop during the travel and/or shipment to the home base before the adequate processing for storage in the gene bank (i.e., an important time factor). Observations on possible selection pressure parameters that the collected material has been or could have been exposed to should always be noted.

From the above, it can be deduced that for a species to be conserved long-term, sampling is critically important. Therefore, due consideration should be given not only to the number of individuals of a population or taxon per site to be collected but also how many populations or sites in the area should be collected and how these should be distributed over the area of a given species in a country or even region to obtain an adequate representation of the “total diversity” present in that area or region. As for most species, detailed information on the distribution of the genetic variation required to decide on the best sampling strategy is missing. One has to extrapolate the information from those species that have been studied to those species where basic information is lacking [9].

A generally accepted benchmark criterion for collecting germplasm is to ensure that at least one copy of 95% of the alleles with a frequency greater than 0.05 is included in the collected sample. Random and unrelated gametes from a population of a target species will meet this criterion. This would be assured by collecting and bulking seeds or vegetative material from 30 randomly chosen individuals in a fully outbreeding sexual species or from 30 random genotypes in an apomictic species or from 59 random individuals in a self-fertilizing species. A sample of 50 individuals from each population and about 50 populations in an ecogeographic area is considered as a benchmark [9,13]. In a later analysis, Hoban concluded that for a metapopulation of wild species, one should collect a minimum of 1000 individuals per metapopulation or an ecogeographic range of the species to compensate for migration between populations and the loss of plants through germination failure, disease, and active use, to preserve enough allele copies to account for various degrees of collection attrition [14].

However, in many instances, the collector is not able to collect such a high number of individuals from a population, as the maturity time of the species might not be optimal, the number of growing plants is limited, etc. and thus the samples collected do not represent the variation of the target species in the area collected. In the case of cultivated crops (which require a different and, in general, more simple collecting strategy), many of the collecting has been carried out for the sake of convenience in marketplaces with all possible implications this might have for the diversity collected, especially for heterogeneous landraces and traditionally mixed materials where sampling the variation in the target materials is not possible and its representation in the market sample likely inadequate. Another limiting factor is that the records taken at the collecting site are limited or sometimes completely lacking, but for the sake of the possibly threatened diversity, such materials are added to the collection [15].

From a gene bank management perspective, a practice of gene bank curators is to collect sufficient seeds and to use the collected material directly for long-term storage and thus to avoid the initial seed multiplication/regeneration. This practice can only work if sufficient seeds of high quality can be collected, a pre-condition that often does not apply [16].

As has been mentioned in the section on the history of conservation, with the entrance into force of the CBD in 1993 in which the sovereignty of states over the (plant) genetic resources in their territories is recognized, a greater hesitation of readily sharing genetic resources with other countries can be observed [17]. In the case of the CGIAR centers, a significant decrease in collecting activities has been observed as several countries and organizations have difficulties providing permission to access genetic diversity for inclusion into the in-trust collections. Some of the reasons for this are uncertainties regarding institutional ownership over genetic resources and unresolved tensions concerning benefit sharing [18].

It should be noted, however, that not all germplasm samples entering gene banks are a result of a collecting mission. In the more recent history of gene banks, many accessions are also obtained as a donation, upon request, from other public or private collection owners (see Figure 1).

### 2.2. Processing

Processing of collected, harvested, regenerated/multiplied, or donated germplasm materials refers to activities during which the materials are being prepared for (long-term) storage or maintenance (in the case of perennial crops kept in field gene banks, in in vitro collections, or tissues and plant propagules cryopreserved). The focus here will be on seed materials as these are the bulk of conserved germplasm. Such preparation steps include the threshing of the seeds from the collected culms; the removal of the seeds from the fruits and where necessary their washing; cleaning or winnowing; the removal of broken, diseased seeds or seeds from different taxa (e.g., weeds), seed drying, packaging, and storage. For details on these various steps, especially on seed drying and other factors that might impact the quality of seed for long-term conservation, see the work of [19] and the Crop Genebank Knowledge Base [20].

Whereas most of the processing steps are straightforward, some aspects that might impact the seed quality for long-term conservation are briefly treated here. In summary, seeds of high quality can be obtained by planting for regeneration/multiplication in suitable areas/fields and at appropriate times; applying suitable crop management practices; adoption of proper harvesting and drying techniques; careful handling and processing to minimize mechanical injuries and unwanted seed mixing with other accessions; and ensuring minimum deterioration before reaching the designated storage, in particular, fast processing and no exposure of seeds to high humidity and temperature. However, seed production and post-harvest handling are highly dependent on the biology and agronomy of the species [19].

While the FAO Genebank Standards [5] recommend seed drying to equilibrium in a controlled environment of 5–20 °C and 10–25% RH and many gene banks operate their drying room at 15 °C and 15% RH, such a drying regime has been found to be not suitable for rice [21,22]. If, due to weather conditions, rice seeds have a high moisture content at the time of harvest (>16.5%), initial drying at 40–45 °C is recommended, followed by final equilibrium drying at 15 °C and 15% RH. This practice has been shown to significantly improve seed longevity during storage compared with standard drying at 15 °C and 15% RH [22]. A similar response was observed with accessions of wild rice [23]. Based on this research, the IRRI gene bank is now routinely using a two-stage drying process for the entire rice collection [6]. Freshly harvested seeds are first dried for three days in a drying room set at 40 °C and 30% RH, followed by equilibrium drying in a drying room set at 15 °C and 15% RH. Some other species might also respond favorably to an initial drying at higher temperatures than 15 °C in terms of seed longevity [22].

One important seed processing step relates to the creation of subsamples of the materials that belong to the same accession with the aim to facilitate “easy access” when material is needed for viability testing, regeneration, or distribution. The idea is that one subsample represents the diversity of the accession adequately and that the number of seeds is meeting the requirements for viability testing (typically 100 seeds, allowing for four replicates of 25 seeds each), for regeneration (typically to ensure that the entire genetic variation is represented in one subsample, i.e., not less than 50 seeds) and/or distribution (typically very limited numbers of seeds). Thus, correct sub-sampling and including sufficient seeds to represent the diversity adequately, especially for long-term conservation, is critical. Of the same importance and nature is the number of individuals used in the regeneration of an accession. Genetic drift is likely to happen when the number of plants is below the effective population size, and thus, genetic erosion might happen in the gene bank.

Another important germplasm processing activity with a possible direct and significant impact on the longevity of the stored seeds is the (timely) drying of the collected/harvested seeds. Whereas one can hardly influence the quality of collected seeds in the field during a collecting mission, many factors can be influenced and optimized to produce high-quality seeds during a regeneration cycle. These factors include cultivation and harvest practices, but also the proper drying of seeds in the gene bank before storage [19]. As the optimum drying is, among others, depending on the species and as the possibility of over-drying has been reported, possibly decreasing the longevity of seeds, it is advisable to conduct straightforward tests to define the optimum seed moisture content for long-term storage [24,25,26].

### 2.3. Seed Longevity

Knowledge of expected seed longevity in storage is important for determining viability monitoring intervals. Seed viability can be predicted with the help of a viability equation developed by Ellis and Roberts [27] for orthodox seeds, using parameters derived from seed storage experiments under different temperature regimes and moisture contents. The Ellis and Roberts viability equation *v = K_i_ − p/σ* shows the relationship between viability and storage period, where *v* is the viability after *p* years in storage, whereby *σ* represents the slope of the curve and *K_i_* the initial viability of the seeds. Meanwhile, improved equations have been developed by Hay et al. [28] and Probert et al. [29]. For a limited number of about 70 species, the seed viability constants can be found in the Seed Information Database (SID) of the Royal Botanic Gardens Kew [30].

Detailed knowledge of crop-specific seed longevity is important for the determination of seed viability testing intervals. The Genebank Standards [5] recommend setting monitoring intervals at “one-third of the time predicted for viability to fall to 85% of initial viability or lower depending on the species or specific accessions, but no longer than 40 years. If this deterioration period cannot be estimated and accessions are being held in long-term storage at −18 °C…, the interval should be 10 years for species expected to be long-lived and five years or less for species expected to be short-lived”. To arrive at reliable seed longevity estimates, analyses of regular viability monitoring data over the entire storage period are essential. However, even among CGIAR gene banks, there is a lack of robust, reliable historical data on the long-term viability of seed lots [31]. Reliable seed longevity estimates would enable gene bank curators to forecast more reliably regeneration requirements, to estimate the size of seed lots required for long-term storage, and to adapt accession monitoring intervals.

USDA seed longevity research has shown that some plant families are characterized by predominantly short-lived seeds (e.g., Apiaceae and Brassicaceae), while others (e.g., Malvaceae and Chenopodiaceae) have long-lived seeds [32]. A meta-analysis of seed longevity studies indicated that seed of maize (*Zea mays*), oat (*Avena sativa*), barley (*Hordeum vulgare*), sorghum (*Sorghum bicolor)*, many grain legumes (*Cicer arietinum*, *Vicia* sp., *Vigna radiata*, *Lens culinaris*, *Phaseolus vulgaris*, *Pisum sativum*, *Trifolium repens*, *Melilotus alba*), and vegetable crops (*Raphanus sativum*, *Abelmoschus esculentus*, *Cucumis melo*, *Cucumis sativus*, *Solanum melogena*, *Solanum lycopersicum*, *Spinacea oleracea*) are long-lived, while seed of rye (*Secale cereale*), groundnut (*Arachis hypgea*), sunflower (*Helianthus annuus*), rapeseed (*Brassica napus*), and some vegetables (*Allium cepa*, *Allium ampeloprasum, Lactuca sativa, Capsicum annuum, Apium graveolens, Daucus carota, Pastinaca sativa*) and some forage grasses tend to be short-lived [32,33]. Surprisingly, the comparison of seed longevity between wild and cultivated species under the same storage conditions did not reveal significant differences. Across all species, the meta-analysis carried out by Solberg et al. [33] indicated a viability loss in the range of 0.2–0.3% per year if seeds were stored under the recommended conditions according to the Genebank Standards [5]. These viability losses are much higher than would be expected by the published viability equations. The multi-faceted aspects and approaches to understanding the inter- and intra-specific differences in seed longevity have been discussed by the global research community during a workshop organized by the International Society for Seed Science (ISSS) in July/August 2018 in Fort Collins, CO, USA, and synthesized by Pritchard [34].

There are many factors that determine initial seed quality and viability and, consequently, have an impact on seed longevity in storage. Among these are crop management practices, climatic factors, stage of seed development at harvest time, and post-harvest seed processing [19]. Differences in geographic origins and, hence, climatic and environmental factors appear to contribute to the variation of P_50_ values within genera and families [32]. Seeds from *Brassica* and *Lolium* species that originated from Europe had characteristically shorter shelf lives than seeds from the same species originating from South Asia and Australia. Ellis [35] stressed that the interaction of genotype with environmental factors determines when maximum seed quality is first attained and for how long it is maintained during the seed development and maturation phase. The period of maximum seed quality may be brief or could be extended depending on several factors. Regarding seed processing factors, research at IRRI has shown that a two-stage drying process significantly improves seed longevity during storage compared to standard drying at 15 °C and 15%, and this modified drying approach has now been adopted for all rice accessions at IRRI [6]. Other species might also respond favorably to an initial drying at higher temperatures than 15 °C in terms of seed longevity [22].

Storage conditions clearly affect seed longevity. Experiments conducted at the CGN gene bank in the Netherlands with seeds of wheat (initial germination rate of 95%) and barley (initial germination rate of 94%), stored at either +4 °C or −20 °C retained high viability of 94% for wheat and 90% for barley after 23–33 years of storage at −20 °C [36]. In contrast, the viability of seeds stored at +5 °C for the same period declined to 62% for wheat and 75% for barley with concomitant losses of seed vigor. A subset of the wheat accessions tested only seven years later showed a further drastic decline in mean germination to 35% when kept at 4 °C, while the samples conserved at −20 °C remained stable at 95%. Similarly, seed longevity studies in maize accessions stored for an average of 48 years at the CIMMYT gene bank in Mexico revealed a significantly lower and more variable seed germination rate of 81.4% for seed lots conserved as an active collection for distribution (at −3 °C), as compared with a high and more stable germination rate of 92.1% of the seed lots conserved as a base collection in a chamber maintained at −15 °C [37]. Based on these long-term storage results of maize accessions, it has been suggested to apply base collection storage conditions (−15 °C) to both the active and base collection to improve seed longevity and reduce the need for costly regeneration events [37].

In a relatively short storage experiment of five years only, no loss in seed viability was detected in any of five species tested during this period when seeds were stored at −20 °C with either low (5.5–6.8%) or ultra-low (2.0–3.7%) seed moisture content [37]. However, significant viability losses were measured after a 5-year storage period at +20 °C, and losses occurred faster at low SMC compared to ultra-low SMC [38].

Molecular approaches to understanding and predicting seed longevity in storage are briefly discussed in Section 4.1.

### 2.4. Seed Regeneration

As gene bank accessions are often collected from a wide range of geographical locations, there is a high probability that original phenotypic variance is lost during ex situ conservation and seed regenerations. This applies more to crop wild relatives than to landraces and commercial cultivars and seems to be caused by selection or gene flow [39]. Multispectral image analysis of seed, i.e., seed phenotyping, has shown to be an effective method for identifying different seed types within a sample of seeds and for verifying whether incoming seeds from a regeneration cycle match the original seeds [40]. While DNA fingerprinting is an effective method to verify the genetic integrity of regenerated seed materials, a complete phenotypic assessment of accessions through high-throughput phenotyping (HTP) during periodic seed regenerations constitutes an alternative option to ensure that original phenotypic features are preserved [41]. The creation of a digital seed file could be the basis for high-speed authentication [42].

HTP tools such as hypospectral imaging have also been successfully used for seed quality, purity, viability, vigor testing, and variety identification in commercial seed lots of various crop species [43,44]. However, these tools can also be used as objective methods for managing gene bank accessions, starting from acquisition to seed regeneration, avoiding physical contamination, and maintaining genetic integrity [40].

### 2.5. Germplasm Exchange

Given the history of crop domestication and global dispersal of crops for food and agriculture, all countries are highly dependent upon plant genetic resources originating from beyond their borders. This dependency has increased over the past 50 years in connection with economic and agricultural development, the globalization of food systems [45], population growth, and climate change. The increasing challenges of crop adaptation to biotic and abiotic stresses exacerbated by climate change and the need to satisfy food and nutrition security of a still-growing global population, reshuffling alleles within a subset of well-performing breeding lines is no longer sufficient to address the global challenges. Plant breeders, therefore, need to broaden the genetic base and introduce specific traits into their breeding populations, and this can be done by resorting to diverse landraces or crop wild relatives that harbor genetic diversity, which was lost during the domestication bottleneck [46].

The international germplasm collections hosted by 11 CGIAR centers include over 760,000 accessions of crops, forages, and trees [47] and constitute a major proportion of the international germplasm exchange. Over the last 10 years (2010–2019), the CGIAR gene banks distributed over 1.1 million PGRFA samples to recipients in 163 countries. During the period from 2017 to 2019, landraces were the most frequently requested materials (50%), followed by breeding materials (24%) and crop wild relatives (13%). Most samples were distributed to advanced research institutes and universities (42%), followed by National Agricultural Research Systems (NARS; 38%), Non-Governmental Organizations (NGOs) and farmers (85), the commercial sector (7%), and others (5%) [47].

Despite major efforts by gene banks to facilitate and enhance the use of the genetic materials conserved in gene banks, these resources are far from being used exhaustively by breeding programs and/or farmers [48]. This is possibly attributable to the scarcity of descriptive information related to accessions conserved in gene banks, the limited use of genomic, phenomic, and information technologies, and, finally, obstacles in implementing national and international policies for benefit sharing [49].

In developing countries, public breeding programs are faced with financial, technical, and policy-related challenges that are limiting the more widespread use of landraces and crop wild relatives [50]. Apart from technical and financial issues, it was especially the lack of a supportive policy environment that was perceived by public sector breeders in developing countries as a major bottleneck restricting their sourcing and use of more diverse genetic resources.

The multilateral system (MLS) established under the ITPGRFA governs the access to the genetic resources of a pool of 64 food and forage crops (referred to as Annex 1 crops to the treaty) under a standard material transfer agreement (SMTA) and the benefit-sharing arising from their use [5]. Many European gene banks also adopted the use of the SMTA for non-Annex 1 crops (for further details on the ITPGRFA, please refer to the work of [3]). Apart from the SMTA, other material transfer agreements (MTAs) are also in use. The Nagoya Protocol regulates access and benefit-sharing under the CBD [5]. In contrast to the SMTA used under the MLS of the ITPGRFA, this is a bilateral agreement between the provider country and germplasm user.

Prior to the shipment of PGRFA (seed, clonal propagules, DNA), the beneficiary needs to sign the SMTA or other MTA. The MTAs regulate the intellectual property rights (IPR) of the requested material and related information, the conditions of its use and distribution to third parties, as well as benefit-sharing arrangements [51]. The SMTA of the International Treaty and most other MTAs only regulate the exchange of physical germplasm materials and do not refer to the exchange of digital sequence information (DSI) or DNA samples extracted from the genetic resources. According to Andersson [52], the following institutions make explicit reference to the exchange of DNA in their MTA: CATIE, Costa Rica; the National Institute of Agrobiological Sciences (NIAS), Japan, the Missouri Botanical Garden, USA; and the Royal Botanic Gardens, Kew, U.K. However, even though some MTAs cover the exchange of DNA samples, there are still different interpretations regarding the question of whether this precludes the patenting of specific genes. In recent years, the governance of digital genomic sequence information has become a contentious issue, and this, in turn, is leading to international disagreement over access and benefit-sharing regulations and blocks the intended expansion of the list of Annex 1 crops [47,53]. The political dimension of DSI is extensively covered in Section 4.4.

Seeds and especially vegetative propagules used for germplasm exchange are known to potentially harbor harmful pathogens, which may lead to transboundary disease spread along with the international movement of germplasm. Quarantine and phytosanitary measures have been adopted by most countries around the globe to minimize the threat of disease spread by screening export and import consignments of germplasm. The effectiveness of these measures depends on seed phytosanitation treatments, the actual knowledge of pathogen distribution and associated risks, the development, adaptation, and availability of diagnostic tools and protocols for seed health testing, qualified operators, procedures for inspection, and post-entry quarantine facilities [54]. Within the CGIAR gene banks, germplasm health units (GHUs) are responsible for germplasm phytosanitation and testing of the health status to guarantee safe global germplasm movement and exchange and the prevention of the transboundary spread of pests and diseases [54]. In their recent review, Kumar et al. [54] describe in detail current procedures for germplasm health testing and pathogen elimination for the major CGIAR mandate crops. As GHUs are widely distributed in developing countries and are known for their high-level expertise and technical capability, they could evolve into a global network of phytosanitary hubs for the research, diagnoses, control of established and emerging pests and their elimination from germplasm propagules, thus, guaranteeing the safe international movement of germplasm.

### 2.6. Documentation

All routine gene bank operations produce data that need to be captured and documented for internal use and, in many instances, for sharing with germplasm users (passport, characterization, and evaluation data). Adequate information management is important for the safe operation of a gene bank. This includes data on the acquisition, registration, storage conditions and collection type (base, active collection, safety duplicate), monitoring of the viability of accessions prior to storage and in storage, regeneration, characterization, and evaluation, germplasm health testing, distribution, and number of sub- samples and seed quantity of each accession kept in the gene bank [5,55]. In addition, among CGIAR gene banks, weaknesses in effective and consistent documentation of routine gene bank operations have recently been revealed [31]. Accession-level data, which are of high relevance for germplasm users, are passport, characterization, evaluation, and lately also omics data [56]. Internationally accepted multi-crop passport descriptors (MCPD) [57] have been adopted by most gene banks as standards for documenting passport, characterization, and evaluation data. These descriptors allow the exchange of accession-level data between gene banks and the operation of international information data portals on PGRFA, such as Genesys, the FAO-led World Information and Early Warning System on Plant Genetic Resources for Food and Agriculture (WIEWS), the European Search Catalog for Plant Genetic Resources (EURISCO), and GRIN-Global [55,58]. As noted by CGIAR gene bank managers, one of the most important factors affecting demand for PGRFA is the quality, comprehensiveness, and relevance of the accession-level information that gene banks compile about the materials in their collections and make accessible online [47]. Although gene banks are encouraged to also record molecular data being generated through genomics, proteomics, metabolomics, phenomics, and bioinformatics [5,56], these are mostly not generated by gene bank staff themselves but through collaborative projects with other research teams or by specialized research institutes. Given their complexity and exponential increase in volume over time, omics data are stored and made accessible in specific public databases or dedicated to other systems [56]. The GenBank platform [59] hosted by the National Center for Biotechnology Information (NCBI) is such a public database for genomics data.

In line with the Plant Treaty’s global information system (GLIS), digital object identifiers (DOIs) have been introduced for gene bank material [60], and the CGIAR gene banks, as well as the Dutch gene bank CGN, have assigned DOIs to almost all their gene bank accessions [47,56]. DOIs provide a globally unique and permanent mechanism for identifying germplasm [61] and derived products such as DNA and related publications, thus assisting the user community as well as facilitating access- and benefit-sharing mechanisms. However, gene banks still face technical challenges, such as how many different DOIs should be assigned to the same accession. As regeneration in the gene bank environment might result in changes of the original material, often collected in far-away places, should; therefore, accessions of each regeneration have their own (new) DOI [62]? Another question relates to different forms of conservation (seed, in vitro culture, cryopreservation, herbarium) and storage conditions (short, medium, and long term). Should the same material conserved under different forms and conditions receive a different DOI? Should purified lines from heterogenous accessions receive a new DOI? There are also challenges with including DOIs in publications to be automatically discovered as this would require that all DOIs must be listed in the reference section of a paper, and this is not the standard practice of journals.

### 2.7. Research

Long-term conservation of plant genetic resources in gene banks aims at maintaining the genetic diversity of individual accessions as authentic and as close as possible to the original genetic composition. In the critical assessment of current practices and the underpinning theories, we have identified possible research topics that gene banks could (or possibly even should) undertake, where relevant or necessary in collaboration with specialists, to contribute to more effective and efficient long-term conservation and/or to rationalize the current global long-term conservation system. Another reason for making such suggestions is to demonstrate that gene banks are not “dead conservation morgues” but lively, dynamic, and essential institutions that use advances in science to continuously improve the knowledge on the conserved materials and on the applied procedures and to make these as cost-efficient and effective to reduce the burden on future generations. A third reason to promote research on the conservation procedures and on the materials conserved is to involve and widen the participation of researchers in the actual conservation of PGRFA efforts, as that is being seen as an essential responsibility by society. Any contribution to make this happening is important and should be duly recognized.

Examples of how (routine) research activities can underpin optimum management of germplasm accessions and collections, and thus, e.g., contribute to extending seed longevity of accessions are summarized in Table 1.

During the 1980s, IBPGR, later denominated IPGRI and now Bioversity International, initiated research on ultra-dry seed storage, based on the assumption that seeds dried to levels well below 5% SMC could be stored at room temperature for extended periods [63]. However, concerns had been raised regarding a possible over-drying of seeds, potentially leading to a loss of longevity [25]. This issue led to a scientific debate and a global seed project to resolve the controversial points [64]. Results of experiments with *Lactuca* seeds showed that crop species have an optimum seed moisture content and that drying seeds below the optimum water content does neither benefit nor damage seed longevity. Furthermore, it was observed that there is a temperature x water content interaction affecting seed longevity and that for storage of seeds at room temperature (about 25 °C), the appropriate RH of the storage room is about 14%. Conducting research on groundnut (*Arachis hypogaea*) seeds, Sastry et al. [65] were able to demonstrate the potential benefit of ultra-dry storage. When groundnut seeds were dried to 1.7% SMC and stored at 50 °C in aluminum foil bags, seeds retained viability up to 192 weeks (3.7 years) under vacuum storage in incubators, compared to only 144 weeks (2.7 years) when stored in normal atmosphere in an incubator. Seed storage at a temperature of 35 °C and at SMC levels of 1.7% or 3.4% retained seed viability even after more than 5 years.

## 3. Strengths and Weaknesses of the Current Active and Base Collection Concept

The term germplasm collection requires some attention as it is a term that clearly refers to the ex situ conservation scenario and that can encompass all the accessions of a given species or crop, a subset of selected accessions (e.g., a core collection), or to all accessions of the various species that make up the gene pool of a given crop. However, it can also refer to all the accessions stored in a gene bank. Thus, some care in using this term is required, also as it is frequently used for the activity of collecting germplasm. Thus, regarding the latter, it is proposed to use the term “collecting”.

In many instances, gene bank collections have grown out of collections established by plant breeders that were eventually “converted” into a germplasm collection. In addition, research collections established at universities often formed the starting point of subsequent gene bank collections. It should be noted that in both cases, the collections were likely not built for long-term conservation of genetic diversity per se, and thus, their origin might less reflect the genetic diversity of collecting areas, as would germplasm collections that have been formed with samples that were collected for that purpose. Traditionally, such breeding or research collections were conserved at cool temperatures (possibly +15 °C or less) and in paper or cloth bags under controlled relative humidity. It was only during the late 1950s that research on storage temperatures started and that a two-tiered conservation strategy was defined: 1. long-term conservation of “base collections” of adequately dried seeds, usually stored at −18 °C in hermetically closed containers; and 2. medium-term storage of “active collections” under less stringent conditions, e.g., +5 °C and controlled relative air humidity of approximately 35%; both collection types were maintained in insulated storage rooms [66].

In the following, we will take a critical look at the various collection types, along with their proposed storage conditions, but with a clear focus on their long-term conservation strategic aspects. The latter comprises the base and the security backup collections that jointly provide the conditions for long-term conservation. The active collection is meant for the storage of samples for characterization and evaluation, multiplication/regeneration, research, and distribution purposes, usually at a higher temperature than the base collection. A fourth collection type is the archive collection that consists of accessions that are meant to be disposed of but that are kept without any management at −18 °C for security purposes only [67].

### 3.1. Active Collection

Accessions that are being conserved for their use in research activities, i.e., characterization and evaluation, molecular studies, and/or for distribution, are kept under conditions that would provide for short- or medium-term duration, i.e., up to 30 years, depending on the species and seed quality. Thus, a well-cooled storage room at plus 5 °C and with controlled humidity at +/−35% RH would provide these conditions and allow storage in paper or net bags, or any other “open” storage forms, and would allow rather easy access (no cold room protection suites, etc.), whereas also the use of hermetically sealed containers can be used. The agreed FAO Genebank Standards define an active accession as “A germplasm accession that is used for regeneration, multiplication, distribution, characterization and evaluation. Active collections are maintained in short to medium term storage and are usually duplicated from a base collection maintained in medium to long term storage.’ Active collection samples for medium-term conditions should be stored under refrigeration at 5–10 °C and relative humidity of 15% ± 3%” [5]. In this early conservation concept, it was assumed that the material would be turned over within the medium term and restocked with newly regenerated germplasm samples from the base collection. This “rough” concept worked well in cases where both the active and base collections of the same materials were stored at the same gene bank, or in cases where a well-developed networked system between base and active collections had been established, e.g., as at NPGRS and the regional centers of the USDA. However, in many other gene banks where the two collection types were split, this concept failed, for a range of reasons, including inadequate refrigeration, the lack of adequate viability testing, and a too strong focus (if any) on the active collection.

Because of varying local conditions with respect to the precise objectives of the gene banks, the availability of adequate infrastructure, as well as of human and financial resources, refinements of this concept should be undertaken to optimize the conservation [67]. Besides the “local fine-tuning” of the traditional concept, there are several specific reasons that would justify a more critical assessment of this traditional concept and possibly a revision of the accepted practices. These reasons have been updated and expanded from Sackville Hamilton et al. [66] (see Table 2).

### 3.2. Base Collection

The objective of the base collection of a given species is to maintain accessions that are distinct, with respect to the genetic integrity as close as possible to the original sample, conserved for the long-term and not intended for distribution [66]. Furthermore, the base collection should contain as much as possible genetic diversity in a rational (i.e., as few as possible accessions), effective and efficient manner under controlled, secured, and safe conditions for the longest possible time. This collection type is strongly focused on long-term conservation but should also consider the facilitation of the use of the conserved materials whenever possible and without compromising the objective of the conservation. According to the FAO Genebank Standards, the agreed conditions are as follows: “Most-original-samples and safety duplicate samples should be stored under long-term conditions (base collections) at a temperature of −18 ± 3 °C and a relative humidity of 15% ± 3%” [5]. If samples conserved under LTS and MTS conditions are kept in hermetically sealed containers (as recommended), then RH control of the storage room would not be required and is common practice in many gene banks [6,55,69].

The most original sample (MOS) is being defined as: A sample of seeds that have undergone the lowest number of regenerations since the material was acquired by the gene bank, as recommended for storage as a base collection. It may be a subsample of the original seed lot or a seed sample from the first regeneration cycle if the original seed lot required regeneration before storage [5]. Furthermore, the MOS should be prepared and stored under the best possible conditions for safe long-term survival; the seed from the MOS should never be distributed for use. The number of seeds in the “primary MOS” (the sample stored at the gene bank for its conservation) should be sufficient to: (a) allow for the optimum regeneration of the MOS (at least the minimum amount to represent the genetic diversity of the original sample and/or the minimum amount needed to reproduce sufficient seeds for the next generation plus a safety margin). The seeds of the primary MOS should not be touched until the viability begins to drop; (b) conduct routine and smart viability tests to determine when the MOS must be regenerated; and (c) supply the seed that is required for regenerating materials for distribution as part of the active collection. It should be considered to allow stock for several regeneration cycles to avoid that the materials of the primary MOS are depleted before it starts losing viability [66]. The storage of the primary MOS samples can be performed in one or in several containers, but all under the same optimal storage conditions.

Besides the primary MOS samples that make up the base collection, a subsample (the “secondary MOS”) should be stored for security reasons under the same or better conditions than the base collection at another distant gene bank to protect the base collection material against accidental loss [20]. It is called the security backup or safety duplication collection and will be maintained under black-box conditions. The latter means that the recipient gene bank has no responsibilities for viability testing and should never use, regenerate, or distribute these safety duplicates without instructions from the duplicating gene bank. The secondary MOS should only be recalled in case of loss of the primary MOS and should contain sufficient seeds for one regeneration cycle. A viability monitoring routine of the primary MOS needs to be established and should be performed in the most efficient way, i.e., to use as few as possible seeds. In case regenerated subsamples for distribution are stored separately in the active collection, they could serve as an indicator for the viability of the primary MOS if stored under the same or less strict conditions (Figure 1) [66].

In another conceptual scenario, the term “base collection” is used to define a set of accessions that are designated to form a base collection of a given crop. These designated accessions can be stored in the respective gene banks that maintain part of the unique diversity of a given species (i.e., each gene bank conserves a fraction of the global genetic diversity of that species), and collectively all gene banks maintain the “global base collection” for that species. Through its role to stimulate and facilitate the collecting of threatened germplasm, IBPGR established a network of gene banks that had formally agreed to maintain germplasm materials collected with the help of IBPGR/IPGRI for a given crop gene pool in their respective base collections for the long term (the Register of Base Collections) [70] and to make this material readily available to bona fide users. The gene banks of the CGIAR are an important part of this register as they hold global collections of their mandate crops and aim to cover an adequate representation of the total diversity in the respective base collections. In many instances, also supportive crop networks have been established to coordinate and implement these efforts with the collaborating national programs. This network concept was developed by IBPGR/IPGRI, also at the regional level, to network national and institutional gene bank programs to strengthen the collaboration between active and base collections. Furthermore, it was felt that not every genebank had to have a base collection and that the collaboration between active and base collections could be organized at the regional level through regional networks. Europe initiated since approximately 2009 a virtual gene bank collection of unique and important accessions, spread across the gene banks of the continent and collectively recorded in the European germplasm database EURISCO [71]. However, possibly except for Europe, these regional PGRFA networks have disappeared or are dysfunctional.

Weaknesses of existing base collections and options to overcome those weaknesses to arrive at a more secure and rational long-term conservation system of PGRFA are summarized in Table 3.

### 3.3. Other Collection Types

Backup collections. Besides the base and active collections, we have referred in the above also to safety duplicate or security backup collections that are arranged based on black-box agreements between different gene banks. These backup collections can consist of subsamples of accessions from the base as well as the active collection. It is important to stress the importance that the storage conditions of the safety duplicates at the recipient gene bank should be the same or better than those at the “conservation” gene bank. Apart from these bilateral arrangements, the Svalbard Global Seed Vault serves as a global long-term seed storage facility to provide an additional security backup to germplasm stored in gene banks around the world. It is built into a permafrost mountain at Svalbard, and the storage temperature is maintained at −18 °C through an additional solar energy-based cooling system that counters the global rise of the earth temperature caused by climate change, which is also witnessed at Svalbard. The seed vault will only agree to receive seeds that are shared under the multilateral system (see above) or under Article 15 of the International Treaty or seeds that have originated in the country of the depositor. The black-box system entails that the depositor is the only one that can withdraw the seeds and open the boxes [73].Archive collections. The archive collection consists of germplasm accessions that are stored under optimal conditions at relatively low cost but that are not actively maintained. The gene bank does not have (anymore) the responsibility for conserving or distributing these accessions. The type of accessions or materials stored in the archive collection could include the following: a. black-box conservation of experimental materials that could have an IPR protection; b. in case a collection has to be disbanded and yet no other gene bank could be identified to accept that collection, the accessions should be temporarily stored in the archive; c. as per some examples mentioned before (e.g., possible duplicates; extra subsamples), in cases when the curator decides to discard accessions they could be archived instead; d. in case of a “forced” rationalization of the collection selected accessions might be removed from the collection, and they should be considered for archiving until a solution is found [66].Research collections. Research collections contain materials that stem from past research activities and have been kept by researchers or their institutes, sometimes over long periods of time. In addition, collections of plant breeding materials could have been stored or maintained by individual breeders, researchers, or institutes. Depending on the type of activities, some collections might contain very specific materials that are difficult to keep and/or to regenerate and might require specialized knowledge. Since the advent of molecular research, the increased importance of DNA materials can be observed but also of single-seed descendent collections of very uniform quality. Whereas the latter might not be important from a genetic diversity perspective, they might well contain important material for molecular and genomic research as they contain the diversity in a suitable form for such research.Structured collections. Another type of collection that stems from research activities on germplasm materials by structuring the collections on the basis of a specific characteristic or trait (e.g., core, mini-core, trait collections) or also to select the genetically unique accessions maintained by gene banks in a regional context (e.g., the AEGIS initiative in Europe) to form a virtual collection are examples of structured collections. They could be virtual or physical collections.Reference collections. Another type of collection that possibly falls somewhat outside the “direct conservation” related objective of a gene bank is the seed reference collection. The concept stems from the botanic garden world as part of the herbarium “system” that was adapted by gene banks to increase the security of the accessions by allowing the detection of possible mistakes, for instance, during regeneration by comparing the phenotypic features of the seeds of a given harvested accession with those stored in the reference collection and comparing the accession number(s).Non-seed collections. In ex situ conservation, other collection types have been created to maintain specific forms of plants, e.g., field gene bank collections in which accessions are being maintained of entire plants for practical reasons such as the need to maintain the genetic constitution of a vegetatively propagated crop (e.g., potato and many other root and tuber crops), or that the seeds are recalcitrant and cannot be dried without killing the seeds (e.g., avocado, cacao, many other especially tropical crops or species); when tissue cultures of plants are maintained in specially equipped rooms the term in vitro collection is used; in case such materials have been cryopreserved by placing them in liquid nitrogen the term cryo-collection is used [3].

### 3.4. Linking Conservation and Use

In the long-term conservation concept, the base collection is at the center of the strategy and comprises accessions that include the most original sample as well as a representative smaller sample deposited at a distant gene bank for safety reasons, all stored under optimum conditions. In addition, regenerated subsamples of the MOS are maintained in the active collection and are intended for research and distribution. Whereas the storage conditions can be less stringent to facilitate access and as the turn-over of materials is assumed to be faster as the loss of viability, this collection type is the joint between conservation and use.

Possibly the biggest hurdle between (long-term) conservation and use is the strong focus on genetic diversity integrity of accessions and their representation of the sampled diversity in a population or landrace, whereas users are predominantly interested in materials that can be easily used and that have ample data on their genetic makeup and agronomic performance. In addition, the ease of use, both in terms of time and of preparatory steps needed before the material can be used, is an important aspect, and this frequently is related to the degree of uniformity of accession or sample. It might be noteworthy, as already mentioned before, that many of the current base collections stem from past breeding collections and, thus, it can be assumed that many of the accessions are relatively uniform. In addition, during the 1980s and 1990s, it was observed that some genebanks had adopted a strategy to remove “off-types” from (landrace) accessions of self-breeding crops, e.g., ICRISAT in its sorghum collection [74]. Thus, we present ideas and examples of how long-term conservation and the facilitation of use can be improved:The creation of core and mini-core collections facilitates germplasm screening and selection for breeding purposes (see Section 4.2). Similarly, the inclusion of specific breeding or discovery populations created through introgressions from the wild into cultivated backgrounds into the active collection will enhance germplasm use in plant breeding (see Section 4.2);Whenever possible, more uniform materials created from a diverse accession could (or possibly should) be kept as separate subsamples in the active collection for distribution purposes. Examples of such “more uniform” subsamples could be pure lines selected and created from genotypes of self-breeding landraces and single-seed descendent lines, prepared for sequence studies. However, such subsamples will always remain part of the active collection and will not become part of the base collection;Possibly the most critical factor that triggers the use of accessions by plant breeders is the availability of comprehensive data on the performance of the accession, on specific traits or characteristics obtained through characterization or evaluation activities, molecular and genomics data, as well as data from genotyping and phenotyping efforts. Thus, a gene bank should generate or facilitate the generation of such information and make the data readily available online as well as through publications. In addition, the availability of comprehensive passport data will be of relevance, for example, for the FIGS approach (see Section 4.2) to see if certain traits or characteristics that are environment-related have a specific and well-defined origin;One aspect that could reduce the “tension” between conservation and use is the economic rationalization of the conservation operation through optimizing the storage conditions for the active collection whenever possible. As already mentioned before, keeping the active collection under suboptimal conditions, i.e., triggering regeneration by loss of viability and not by depletion of the stock, is a real cost factor as well as a potential threat to the integrity of the genetic diversity of an accession.

## 4. New Developments That Facilitate More Effective and Efficient Conservation and Use of PGRFA

Major technological advances, particularly in DNA sequencing, molecular biology, and omics technologies, phenomics (including sensors, imaging, robotics,) computation, information science, and the management of big data enable a transformation of the way in which plant genetic resources are managed and used. These advances are attracting a considerable number of new clients to gene banks, such as molecular biologists and geneticists alongside molecular and traditional plant breeders and may affect the operations of gene banks and aspects of their future role [75,76]. It is a major challenge for gene banks to satisfy the needs of this wide range of users, each group with a different set of expectations.

### 4.1. Role of Molecular Biology and Genomics in Promoting Long-Term Conservation

New applications of modern molecular biology tools and techniques such as next-generation sequencing (NGS) and genotyping-by-sequencing (GBS) enable scientists to enhance the quality, efficiency, and cost-effectiveness of gene bank operations, as well as the depth of scientific knowledge of gene bank holdings, thereby also guiding conservation management [77,78,79].

#### 4.1.1. Redundancy in Crop Collections

Molecular tools are certainly helpful for making informed decisions on reducing redundancy in crop collections, thus contributing to efficient long-term conservation [80,81,82]. However, this approach is not straightforward. Given the fact that even in self-pollinating species within accession variation may be considerable, there is a need to statistically quantify variation within and among accessions to decide whether they are sufficiently different to consider them distinct accessions [80]. However, from a user perspective, the functional diversity of a single trait, such as disease resistance, is often of high relevance. Given the uniformity of modern cultivars, a certain accession/cultivar may appear redundant based on molecular data but nevertheless differ in a single important trait that is highly relevant for a breeder.

Verification of passport data is the first element that might provide a clue on potential duplicates in a crop collection. This suspicion needs to be verified by morphological comparison of accessions followed by molecular marker techniques that can detect genetic differences between and within accessions [83]. Combining phenotyping and genotyping with single sequence repeats (SSR) markers allowed the identification of duplicate accessions in lettuce (*Lactuca sativa*) and the determination of the most appropriate accessions (MAA) for inclusion into AEGIS [84]. Similarly, the combination of morphological characterization with genotyping using an SNP (single nucleotide polymorphism) array, originally developed for *Brassica napus,* allowed the identification of duplicate accessions in *Brassica oleracea,* and a subset of 500 SNP markers have been suggested for genotyping *Brassica oleracea* accessions [85].

However, the correct identification of accession duplicates within a given crop collection across different institutes is challenging as collaborating institutes have to agree on a common, crop-specific set of markers and, subsequently, there might be problems to reliably reproduce DNA marker data between different laboratories [78]. Such difficulties could be overcome by using next-generation sequencing platforms to tackle the issue of redundancy within and between crop collections of different holding institutes [77,79,86,87]. In a case study comprising three gene banks (CIMMYT; the Wheat Genetics Resource Center (WGRC) at Kansas State University in Manhattan, KS, USA; the Punjab Agricultural University (PAU), Ludhiana, India) and focusing on *Aegilops tauschii*, a wild crop relative of wheat and source of genetic diversity for wheat improvement, Singh et al. [87] identified and characterized over 50% duplicated accessions on average within gene banks. With increasingly more powerful tools to compare genetic information between individuals/accessions, the likelihood of finding two absolute duplicates will decrease. Therefore, in deciding whether two accessions are duplicates, it is advisable to use phenotypic and genetic data for the comparison and to take a decision thereafter.

Genebank scientists of the International Rice Research Institute (IRRI) have pioneered the incorporation of genomics-based research into gene bank activities. IRRI scientists use such data to classify the degree of genetic similarity between accessions and the diversity within accessions to shed light on population structure and admixture, to classify potential duplicates, and to identify genetic novelty [77]. Genomic information will also provide a rationale for avoiding redundancies, thus limiting the size of collections, as well as facilitate genetic gap analyses to guide future collecting and acquisitions. Before incorporating new accessions into a gene bank collection, sequencing data can also be used to make an informed decision whether the new material possesses genuine genetic novelty to deserve inclusion into the base and active collections of the gene bank [77]

#### 4.1.2. Inferring Missing Passport Data 

Often, vital metadata, such as geographical data or taxonomic information on the species of accessions conserved in gene banks, is missing or incorrect. Curating such data is important as accessions with incomplete passport data and missing associated metadata are rarely requested by users [88]. A combination of existing genomic tools and statistical analyses can be used to infer missing pieces such as geographical region of origin as shown by Singh et al. [87] with accessions of *Aegilops tauschii*, a wild relative of wheat.

#### 4.1.3. Predicting Seed Longevity 

DNA protection and repair are important for maintaining genome integrity and seed longevity in plants. DNA damage in stored seeds results in faulty transcription and replication, thus, affecting key processes that are activated during the imbibition stage of seed germination [89,90]. Telomere lengthening has been proposed as a tool to distinguish between short- and long-lived species [90]. Telomere lengthening occurs after seed imbibition when metabolic activities resume, whereas telomere degradation is associated with seed aging. Reduction in seed longevity is often associated with the oxidation of cellular constituents such as nucleic acids, proteins, and lipids [91].

Seeds possess protective mechanisms to prevent damage to their cellular constituents through the formation of glassy cytoplasm that reduces cellular metabolism and the production of antioxidants that prevent the accumulation of oxidized macromolecules during seed storage [91]. Moreover, seeds also have repair mechanisms that remove damage in DNA, RNA, and proteins that accumulate during seed storage. This repair mechanism sets in during seed imbibition through the activation of enzymes such as DNA glycosylase and methionine sulfoxide reductase [89,91]. Through genome-wide association (GWA) analysis in diverse Indica rice varieties, eight major loci associated with seed longevity parameters were identified [92]. Based on their research, Lee et al. [92] concluded that high seed longevity in rice might be related to DNA repair and transcription mechanisms, sugar metabolism, reactive oxygen species scavenging, and embryonic/root development.

A complex network of putative longevity-related genes has been reported by Righetti et al. [93] that links seed longevity to biotic defense-related pathways. Genotypic variation of seed longevity in storage might be determined by two sets of genes [34]. A major set of genes evolved specifically for storability, while the other set is linked to seed dormancy. Metabolomics is a complementary approach to dissect the complexity of seed longevity. A shorter-lived rice cultivar (IIT998) showed a 2- to 6-fold increase in the change of sugar-related metabolites and glutathione-related proteins during natural seed aging compared with another cultivar (BY998) with extended seed longevity [34]. The rapidly increasing availability of reference genomes and pan-genomes constitutes another approach to dissecting and understanding the complexity of seed longevity [34].

Experiments with seeds of several vegetable crops have shown that RNA integrity declines with storage time in dry seeds [94]. As a decrease in RNA integrity was usually observed before viability loss, this assessment can be used to predict the onset of viability decline. Observing DNA and RNA integrity loss and understanding repair pathways in stored seed could help predict seed longevity and determine seed viability testing intervals. This information could also be used to develop crop varieties with improved seed storability and enhanced germination performance [89,95].

### 4.2. Role of Functional Genomics and Phenomics in Facilitating the Use of Plant Genetic Resources Conserved in Genebanks

Different strategies have been developed to select and prioritize potentially useful accessions from gene banks that can be used for crop improvement. Among those is the development of core or mini-core collections. Frankel [96] coined the term “core collection” meant to “represent with a minimum of repetitiveness, the genetic diversity of a crop species and its relatives”. A core collection is a subset of a large collection, consisting of about 10% of the accessions and capturing most of the genetic diversity available in the entire collection [97]. As core collections of large crop collections such as those maintained by CGIAR gene banks are still too large for use by breeders, Upadhyaya and Ortiz [98] developed a mini-core collection concept, which is based on the evaluation and selection of a further subset of about 10% accessions from an existing core. ICRISAT has sent mini-core collections of chickpea, groundnut, pigeonpea, sorghum, pearl millet, foxtail millet, and finger millet to different research groups in 14 countries [99]. The World Vegetable Center has developed mungbean core and mini-core collections [100] and even a core collection of the wild tomato species *Solanum pimpinellifolium* [101] and offers these special collections together with the respective accession-level phenotypic and genotypic data for distribution to interested researchers and breeders to enhance the access to biodiverse vegetable germplasm for breeding and research.

Another strategy of selecting accessions based on their phenotype and associated passport data is the focused identification of germplasm strategy (FIGS) that is based on the assumption that the adaptive traits expressed by accessions are the direct result of environmental conditions of their respective place of origin and that the genetic diversity of the specific traits of interest can be maximized by sampling accessions based on their diverse contrasting geographic regions [102,103]. However, accessions conserved in gene banks around the globe are often lacking phenotypic data, and passport data might also be incomplete or have incorrect location data, which limits the application of FIGS. Hence other methods are required to facilitate the use of germplasm for breeding.

With the advances in molecular biology and genomics, DNA extracted from nuclei, mitochondria and chloroplasts are increasingly being used to evaluate patterns of genetic variation within and among species, map and characterize desirable traits and underlying genes of interest for breeding, for taxonomic studies telling species apart [104] and to infer the evolution of genome structure in plants [105,106]. Genotyping-by-sequencing (GBS) combines genotyping and genome-wide molecular marker discovery in one and the same process [79]. It facilitates the exploration of new germplasm sets or species that have not yet been characterized without the need to first discover and characterize polymorphisms through molecular marker studies. Moreover, with the advancements of bioinformatics, the development of new reference genomes, and the availability of an increased volume of sequence data, GBS data sets can later be reanalyzed to uncover further information, such as new polymorphisms or annotated genes.

Large crop collections cannot be sequenced in one go. There is a need to develop or use existing core collections and to transform them into genetic stocks for reference purposes and comparative and integrative genomic studies [107]. Heterogenous accessions must be purified through single-seed descent (SSD) before DNA extraction and initiating systematic molecular characterization. The purity of accessions can be assessed through genotyping with various molecular marker types such as inter-simple sequence repeats (ISSRs) or amplified fragment length polymorphism (AFLPs) [78].

These purified accessions or lines will also serve as the source material for phenotyping and ensure that phenotypic information can be properly linked with the sequence information in a meaningful way [77]. As has been demonstrated for rice, the demand for core subsets may substantially increase if all the accessions in the subset have been sequenced, and this information is made easily accessible to breeders or other researchers [42]. This information allows users to perform their own genome-wide association studies to elucidate the genetic control of multiple traits of interest. Therefore, this concept should be applied to all major crop collections conserved by a gene bank. DNA genotyping and sequencing results in combination with precise phenotyping are perfect assets for trait mapping, gene analysis, and allele mining in support of modern plant breeding.

An important objective of functional genomics in agricultural species is the use of sequence polymorphisms for phenotypic predictions and the selection of improved plant types. Prediction models are built by correlating phenotype and genotype in a breeding population of interest, and these models allow the identification of individuals with superior breeding values [108,109]. The suitability of GBS markers in developing genomic selection models has been verified in the complex wheat genome with the prediction for yield and other agronomic traits [108]. In rice inbred lines, genomic prediction models outperformed prediction based on pedigree records alone for three traits, i.e., grain yield, plant height, and flowering time [110]. Meanwhile, genomic selection has been recognized as an excellent tool to estimate genomic breeding values and is widely used in crop breeding [111].

Even if genetic stocks required for GWS and genomic prediction do not contain unique genetic novelty and, therefore, do not merit long-term conservation, they constitute important assets for genomic research. They could be kept together with specific breeding populations in the active collection under medium-term storage conditions to support future research and breeding needs [77]. Such specific breeding or discovery populations created through introgressiomics [112] would include collections of recombinant inbred lines (RILs), backcross introgression lines (BILs), chromosome segment substitution lines (CSSLs), multiparent advanced generation intercross (MAGIC) lines, nested association mapping (NAM) populations [113] and other training populations developed to represent different breeding pools [77]. It should be noted that many of these genetic stocks are very difficult to regenerate and that it might require the involvement of the breeders concerned to assist in such efforts.

With the current advances in NGS and GBS, the growing volume of fully annotated genomes, and knowledge of candidate genes, genotypic accession-level data is no longer of major concern [76]. The lack of high-quality phenotypic data is currently the major bottleneck for functional genomics and the efficient exploitation and use of germplasm accessions in modern breeding. With the advances in high-throughput phenotyping (HTP), there should now be a major focus on phenomics in crop collections to complement genotyping data. Phenotyping is an expensive yet indispensable component of plant research and crop improvement programs that helps to understand the genetic basis of traits and the interaction between genotypes and the environment [41]. Phenomics aims at bridging the gap between genomics, plant function, and agricultural traits [114]. While “forward phenomics” uses phenotyping tools to “sieve” collections of germplasm for valuable traits, such as yield components, biotic or abiotic stresses, “reverse phenomics” dissects those traits to reveal underlying mechanisms, such as biochemical or biophysical processes and ultimately the gene(s) regulating those processes [114].

Genebank phenomics is a novel approach in modern gene banking, and Nguyen and Norton [41] shed light on new HTP methods that enable capturing traits during seed regeneration events. One of the valuable features of HTP is that multiple sensors can be deployed at the same time to simultaneously and non-destructively capture several independent observations, which would not be possible through manual observations and measurements as practiced until recently in most gene banks. HTP will allow for more targeted prioritization of accessions from large crop collections for further downstream studies and identification of traits of interest for breeding. Seed phenomics has also been shown to aid genomic prediction for seed traits in barley breeding lines [115]. A convention on the minimum information about plant phenotyping experiments (MIAPPE) has been recommended by the plant phenomics community to ensure easy and correct interpretation, assessment, review, and reproducibility of published data [116].

However, the majority of gene banks might not be able to afford the investment needed for setting up and operating a phenotyping platform. Imaging costs include the imaging hardware, the cost of the vector (e.g., manual measurements, drones, hand-held or automated/robotized ground vehicles), and associated software/pipelines for data capture, storage, organization, and analysis [117]. The latter may represent 30–200% of the cost of image capture, a considerable cost factor of phenotyping, and might best be achieved through research consortia.

Nguyen and Norton [41] proposed a strategic phenomics approach to benefit the management of gene bank collections and to enhance the value and use of PGRFA. If possible, seed regeneration blocks should be replicated with a reasonable number of individuals to facilitate statistical analysis and ensure that the sample size is sufficient to maintain the genetic diversity and integrity of accessions. Using HTP from routine seed regeneration events over subsequent years, an enormous volume of morphological, agronomic, physiological, and environmental data [118] can be collected simultaneously. As most quantitative traits, such as grain yield, cannot be assessed in small regeneration plots, the measurement of secondary correlated traits such as early vigor, height, canopy properties, and biomass during the growth phase may serve as indirect indicators of grain yield and can be used together with GBS data for phenomic and genomic selection from diverse landrace accessions [119,120,121]. The described strategic gene bank phenomics approach has been implemented, for example, in the Australian grains gene bank (AGG) using different HTP platforms [41].

With the advances in biotechnology, the term “synthetic biology” has been created, which may be described as combining functional elements in novel configurations to modify existing properties or to create new ones [122]. Synthetic biology comprises a variety of techniques ranging from systems biology, metabolic engineering (“Golden Rice”), protein engineering, and genetic engineering and is a topic of regulatory concern regarding the biosafety of new products that could potentially fall into the category of living modified organisms (LMO). In addition, the use of digital sequence information (DSI) derived from germplasm is of concern regarding the third objective of the CBD on access to genetic resources and benefit-sharing and the recently concluded Nagoya Protocol [123]. These regulatory concerns are discussed further below in Section 4.4.

Synthetic biology encompasses genome editing and the respective enabling tools such as the “CRISPR” (clustered regularly interspaced short palindromic repeats) technology [124], and applications of CRISPR such as organisms containing engineered gene drives, a genetic strategy to control populations of disease-vectoring insects [125]. Furthermore, synthetic biology also allows de novo domestication of species, such as wild *Solanum pimpinellifolium* with enhanced fruit size, number, and nutritional value of the fruits [126], and multiplex editing, i.e., the simultaneous targeting of several genes with a single molecular construct [126,127].

The discovery of sequence-specific nucleases, including zinc-finger nucleases (ZFNs), transcription activator-like effector nucleases (TALEN), and the CRISPR-Cas system [128], enabled targeted genome editing in a precise and predictable manner in transformable plants by inducing a DNA double-strand break (DSB) at a target site [129]. Thereafter, either the non-homologous end joining (NHEJ) pathway or the donor-dependent homology-directed repair (HDR) pathway repairs the DSB, thereby introducing genetic changes [128,130], which might take the form of gene knockouts or gene replacements.

Gene knockouts are useful to eliminate genes that are detrimental to food quality or that confer susceptibility to plant pathogens [128]. For example, CRISPR/Cas-9 targeted mutagenesis has been used to knock out the powdery mildew susceptibility gene *PMR4* in tomatoes, resulting in enhanced resistance against this pathogen [131]. Wang et al. [132] used both TALEN and CRISPR/Cas9 technologies to target the genes of the mildew-resistance locus (MLO) in hexaploid bread wheat and successfully knocked out all three MLO homoeo-alleles resulting in heritable, broad-spectrum resistance to powdery mildew. The CRISPR–Cas9 system has also shown the potential to directly target plant-infecting gemini viruses by inhibiting virus replication and, thus, enhancing plant resistance to those virus diseases [127,133,134].

While the NHEJ pathway for DSB repair is error-prone and the HDR pathway has a low editing efficiency, newly developed precise CRISPR-Cas technologies rely on deaminase-mediated base editing and reverse transcriptase-mediated prime editing that do not induce DSB formation and do not require donor DNA [130]. These newly developed technologies allow precise nucleotide sequence editing, are more efficient than HDR in plant genome modifications, and show great promise for rapid plant improvement. CRISPR-Cas technologies can work alone or can be combined with conventional breeding methods, thus, accelerating the breeding progress. This has been demonstrated for haploid induction in wheat, maize, and rice, for generating male sterile lines (wheat, tomato), inducing apomixis for fixation of hybrid vigor, for overcoming and restoring incompatibility, and crosses among distant gene pools [130].

The value of underutilized species and wild food plants for food and nutrition security and crop diversification aiming at more sustainable production systems has been demonstrated (see, for example, the work of [135,136]). CRISPR-Cas technologies, with their capacity for precise genome editing, could be used to accelerate the domestication and breeding process of such underutilized crops. For example, the wild tomato species *Solanum pimpinellifolium* shows salt tolerance as well as resistance to fungal and bacterial pathogens [101,137]. Using a multiplex CRISPR–Cas9 strategy to edit genes related to unsatisfactory traits in the wild form such as day-length sensitivity, shoot architecture, flower and fruit production, and nutrient content, Li et al. [138] were able to accelerate the domestication process of *S. pimpinellifolium* without compromising its abiotic and biotic stress tolerance traits. Similarly, the orphan solanaceous crop ground cherry, also called husk tomato (*Physalis pruinosa*), was partially domesticated by mutating orthologous domestication genes of tomato, resulting in plants that were shorter and had more flowers and larger fruits [139]. This is clear evidence that knowledge from model crops such as tomatoes can be used to edit genes to improve agronomic traits of distantly related underutilized crops.

According to Gao [140], the use of CRISPR technology in plant breeding could simply be considered as “a new breeding method that can produce identical results to conventional methods in a much more predictable, faster and even cheaper manner”. Seeing genome editing with CRISPR technology in such a way could eventually help overcome the current prohibition of using this technology in the European Union, where the resulting products are still defined as “GMOs” even if no foreign DNA is introduced.

### 4.3. Specialized Databases, Portals and Networks for Genomics and Phenomics Data Related to Plant Agrobiodiversity

DNA or digital sequences in themselves are of no real value in the absence of information about the samples they were derived from [141]. It is crucial that a collection of plant DNA extracts is intricately linked to the original plant material (and associated information) from which the genomic DNA was derived. To manage the huge amounts of genetic and phenotypic data and make this and other valuable information pertaining to germplasm accessions available for users around the globe, a cooperative platform for data collection, analysis and sharing is required [41,42,142]. Information networks need to be unified, globally accessible, and updated as new research results become available [143]. For this to succeed, the gene bank community will need to link with other information specialists to build a truly global information system with a searchable interface [42]. With the Plant Treaty’s global information system (GLIS), introducing digital object identifiers (DOIs) for gene bank material [60], a first step has been made in that direction. The DOIs assigned to germplasm accessions allow the storage of omics data in specialized databases without losing the link to the original accession from where it was derived from.

Several components and international initiatives already exist to develop different aspects of the required information infrastructure and are listed below. The first three are widely used in the gene bank and germplasm user community, while other listed resources try to better connect the user community with gene banks or with phenomics and genomics data portals.

Genesys, a global platform on PGRFA with free online search engines, provides access to passport and characterization data on accessions conserved in gene banks worldwide [144];GRIN-Global [145], the global germplasm resource information network, provides a scalable version of the USDA-ARS Germplasm Resource Information Network (GRIN) that is suitable for use by any interested gene bank around the world;EURISCO [146], the European plant genetic resources search catalog, receives data from the European National Inventories (NI) and provides accession-level information of PGR conserved in European gene banks or other collections;The DivSeek network, founded in 2012, aims at catalyzing the advanced conservation, management, and traceability of PGRFA through a collaborative network of gene banks, breeders, plant and crop scientists, and database and computational experts [49]. To achieve the goal of value addition to germplasm conserved in gene banks, DivSeek has assembled three working groups that are focusing on genomics, phenomics, and policy;The Breeding API (BrAPI), an interface for exchanging plant phenotype and genotype data between crop breeding applications [147];The Research Data Alliance (RDA), which aims at enabling data sharing, exchange, and interoperability [148];The Global Open Data for Agriculture and Nutrition (GODAN) with the objective of making agricultural and nutritional data available, accessible, usable, and unrestricted [149];The Global Biodiversity Information Facility (GBIF) [150] was established in 2001 in Copenhagen, Denmark, based on a recommendation from the Organization for Economic Cooperation and Development (OECD) Global Science Forum as an international mechanism to promote standardization and aggregation of biodiversity data and (updated) information and make it accessible worldwide. As of 8 May 2021, GBIF had registered an amazing 1.7 billion occurrence records. However, despite this impressive number, only about 21% of preserved collections are digitally accessible via GBIF [151];The Global Genome Biodiversity Network (GGBN) [152], created in 2011 [153], links through its data portal globally distributed biodiversity databases of genomic samples, ensures easy access to DNA and/or tissue samples, and bridges the gap between biodiversity repositories, sequence databases and research results [154]. Within GGBN, a pilot project called GGI-Gardens focuses on the approximately A total of 460 vascular plant families [153]. Under CGBN, access is governed through standard material transfer agreements in compliance with regulations of the Convention on Biological Diversity [155], the Nagoya Protocol on Access and Benefit Sharing [156], and the Convention on International Trade in Endangered Species of Wild Flora and Fauna (CITES) [157];The International Nucleotide Sequence Database Collaboration (INSDC) [158], comprising the DNA DataBank of Japan (DDBJ), the European Nucleotide Archive (ENA), and GenBank at the National Center for Biotechnology Information (NCBI), an annotated collection of all publicly available DNA sequences. It is an archival database that rarely provides updates about specimen or locality data;The International Plant Phenotyping Network (IPPN) [159] represents plant phenotyping centers globally, runs multidisciplinary working groups, and facilitates the sharing of up-to-date information about new HTP infrastructures and methodologies for various crop phenotypes [41,160]. Several regional and national partners are associated with IPPN, two of which are mentioned below;The North American Plant Phenotyping Network (NAPPN) [161] brings together scientists in the evolving area of plant phenomics and is a regional partner of IPPN;EMPHASIS, also a regional partner of IPPN, enables researchers to use facilities, resources, and services for plant phenotyping across Europe [162].

There are also crop-specific consortia such as the International Rice Informatics Consortium (IRIC) [163], the International Wheat Improvement Network (IWIN) [164], the APSA/WorldVeg Vegetable Breeding Consortium [165], and many others.

Taxonomic and evolutionary studies make increasingly use of information provided through data portals or virtual collections where specimen data and images are made available through the Internet. To annotate virtual specimens, online annotation tools are required. A generic online annotation system called *AnnoSys* [166] has recently been developed to access collection data from both conventional web resources or the Biological Collection Access Service (BioCASe) and accepts XML-based data standards such as Access to Biological Collection Data (ABCD) [167] or DarwinCore [168] for data exchange [169]. GBIF and other biodiversity portals are already integrating AnnoSys. Filter types for specimen records in queries and for notifications include: family, genus, species, collector name, collector’s number, country, institution code, collection code, catalog number, identified by, and annotator.

### 4.4. The Political Dimension of Digital Sequence Information (DSI) Sharing

Progress in life sciences, including health, biodiversity protection, and working toward reaching the sustainable development goals is relying on open access to sequence data provided by public sequence databases. A clear example is the current pandemic caused by SARS-CoV-2. Without the rapid sharing of pathogen genetic resources and digital sequence information (DSI) (called genetic sequence data (GSD)) by the World Health Organization (WHO), it would not have been possible to create effective vaccines within a truly short timeframe of under a year. While rapid progress by the scientific community relies on openness and public availability of genetic sequences, fears have been expressed regarding the increasing ease with which genetic material can be transformed into digital information, transmitted, reproduced, and manipulated through advances in sequencing technologies, genome editing and synthetic biology [170,171]. Progress in synthetic biology might soon enable de novo biological design [170], thus, confirming fears of the dematerialization of genetic resources, i.e., making physical access superfluous and in that way threatening the principles of ABS as established under the International Treaty and the Nagoya Protocol of the CBD [171].

Free-for-all access to sequence information (and associated germplasm) is considered by biodiverse countries to be mainly beneficial for user countries and the biotechnology industry and is seen as counterproductive for provider countries, their local communities, and indigenous people who are the custodians of plant agrobiodiversity and who will not be able to benefit if access to DSI is not subject to ABS regulations of prior informed consent (PIC) and mutually agreed terms (MAT) under the Nagoya Protocol of the CBD [172]. Discussions on the current and future access to digital sequence information are currently ongoing under the Plant Treaty, CBD, Nagoya Protocol, and the multilateral Prepared Influenza Preparedness (PIP) Framework as reviewed by Lawson et al. [173]. The United Nations Convention on the Law of the Sea (UNCLOS), which governs international waters and the deep sea, is also developing a new multilateral treaty with the aim to enhance the conservation and sustainable use of marine biological diversity in areas beyond national jurisdiction [174]. Best practices have been outlined to regulate access to marine genetic resources and sequencing data while sharing the benefits derived from such access, a process that is meant to support science and society.

For the time being, DSI is still used as a placeholder, lacking a precise and generally agreed definition [175]. According to Houssen et al. [176], the term DSI may comprise four groups of information:(i)Narrow, covering DNA and RNA only;(ii)Intermediate, including DNA, RNA, and proteins;(iii)Intermediate, including DNA, RNA, proteins, and metabolites;(iv)Broad, including DNA, RNA, protein, metabolites, and traditional knowledge, ecological interactions, etc.

The unsolved definition of DSI is of major concern for gene banks as they share germplasm with associated accession-level information. Should a broad definition of DSI (group iv of Houssen et al. [176]) be adopted, this would seriously affect the disclosure of passport and descriptive data of accessions, and in consequence, would have a major impact on the distribution of germplasm to users [15]. A narrower definition of DSI with restricted access to that information would be less harmful. Should more countries include DSI in the national ABS legislation, gene banks may no longer wish to conserve material from those countries due to increased complexity of germplasm handling and distribution. As has been shown by the current use of an SMTA under the MLS of the ITPGRFA for germplasm sharing, a multilateral approach to the DSI issue would be the best solution; hence, the incorporation of DSI into the SMTA of the ITPGRFA has been proposed to facilitate the smooth operation of gene banks including germplasm distribution in the future [15]. A further open question is whether ABS regulation would only apply to DSI acquired after entry into force of this regulation or would be applied retroactively.

While the next UN Biodiversity Conference of the Parties (CBD COP15) to be held in October 2021 in Kunming, China will discuss the DSI topic, some countries went ahead and have already included DSI interpretations in their national ABS legislation, thus creating legal uncertainties for scientists accessing publicly available DSI. In fact, 15 countries have already adopted legislation on DSI, and 18 more are planning to do so [171]. If a country decides to include DSI in its national ABS regulation, access to and use of DSI derived from their genetic resources may not be free anymore. This means that PIC and MAT would apply to the use of DSI as well in case access to genetic resources is governed by the Nagoya Protocol.

To solve this contentious issue, Lawson et al. [173] proposed two options: (i) a risk framework matrix for valuing information as part of the ABS transaction by attributing an estimated worth to a particular kind of information; (ii) a charge, tax, or levy that would allow externalizing the costs so that information would remain available to be disclosed and exchanged in support of the scientific community. Based on the matrix, passport data on accessions would be considered as of low value, without restrictions (public domain data), while descriptive (phenotypic) data would be treated as restricted public access data, and sequence data would have a time limit restriction with the requirement for reporting the results obtained to the germplasm/DNA provider. The tax or levy might need to be paid by the party accessing the resources (similar to the PIP Framework) or as a levy on contracting parties, similar to the Norway seed sales tax under the MLS of the International Treaty. For 12 consecutive years, Norway has made an annual contribution to the benefit-sharing fund of the International Treaty, an amount that is equivalent to 0.1 per cent of the value of seed and plant material traded in agriculture in Norway every year [177]. The tax or levy option avoids the high ABS transaction costs required to negotiate the value of information in every single transaction and allows the scientific community to disclose and share the generated information freely [173].

The International Nucleotide Sequence Database Collaboration (INSDC) is the central foundation for global sequence information as it connects over 1700 scientific databases and platforms [178]. The INSDC provides the free core infrastructure for DSI deposition, preservation, and global dissemination as part of a scientific collaboration between the European Molecular Biology Laboratory (EMBL; inter-governmental treaty organization), the National Center for Biotechnology Information (NCBI); USA), which is hosting GenBank, and the DNA Databank of Japan (DDBJ) [179]. Rhoden and Scholz [171] noted that tracking and tracing the movement of nucleotide sequence data (NSD) in GenBank, the largest public database platform, is challenging and that scientists in every country of the world are accessing this platform for their work. Therefore, any financial or administrative burden for accessing NSD will affect all scientists worldwide and limit their ability to undertake research and collaborate. A recently published white paper formulated five different policy options under which DSI could be governed, generate revenues for benefit sharing while preserving the current open access system for scientific discovery and publication [178]. Similar options have been proposed by Oldham [141].

Scholz et al. [178] noted historical parallels between CGIAR gene banks that became part of the MLS of the International Treaty about 25 years after the creation of the CGIAR institutes and the INSDC, established in the early 1980s, which now has assembled DSI from every country, continent, ocean, and region in the world and its databases are accessed by users in every country in the world. There is hope that a multilateral perspective, or even a multilateral mechanism that covers multiple international organizations/agreements, such as the CBD, the International Treaty with its MLS, WHO with its PIP Framework, and UNICLOS, could be a practical way forward to solve the contentious DSI issue. From the foregoing, it becomes apparent that the scientific community requires a multilateral, universal framework for accessing DSI if it is to thrive and to contribute to solving current and future global challenges, similar to the situation of PGRFA.

Despite the growing concern of reducing germplasm collections to dematerialized and digitized genomic sequences, the conservation of physical specimens (seed, tissue, living plants) will retain value for future research beyond the DNA code, although we may not be able to anticipate and specify those values right now [180].

## 5. Strengths, Limitations and Opportunities of the Existing Regulatory Framework, the International Network of Ex Situ Base Collections and the Routine Genebank Operations for Genebanks to Effectively Participate in and Contribute to Regional or Global Long-Term Conservation Efforts

Having described the global long-term conservation and exchange/use system above, in this section, we will take a closer look at the existing policy framework in which the global system is embedded (for a detailed description of the policy framework, please see the previous paper [3]). We provide an assessment of the strengths and weaknesses as well as of possible limitations of this framework that exist from the perspective of national, regional, and global gene banks regarding their ability to participate in the regional/global system. We will conduct a similar assessment of possible strengths, weaknesses, and limitations that characterize the current international network of ex situ base collections with a view on the possible participation of gene banks in that network. In the second part of this section, we will critically assess the strengths, weaknesses, and limitations of routine gene bank operations that have an impact on the effective and efficient participation of national, regional, and global gene banks in the global system.

### 5.1. Regulatory Strategic Framework

The Global System for the Conservation and Sustainable Use of Plant Genetic Resources is a system that evolved in and is being managed and coordinated by FAO, under the oversight of the commission, with the aim to ensure safe conservation and to promote the availability and sustainable use of PGRFA. The global system was also an element of the International Undertaking on Plant Genetic Resources. The latter was adopted in 1983 and entirely devoted to the conservation and facilitation of use [181]. More details on this global system are provided in Engels and Ebert [3]. This global system is largely based on the national PGRFA programs around the world, the botanic gardens maintaining PGRFA, the gene banks of the regional and international research centers of the CGIAR and AIRCA (Association of International Research and Development Centers for Agriculture, among which the World Vegetable Center and ICBA [International Center for Biosaline Agriculture] maintain considerable germplasm resources), the regional PGRFA networks, the global crop networks and other loosely related institutions that are concerned with the conservation and research of biodiversity.

As in any global “project”, it is important to have a well-defined strategy that determines the scope (the “what”), procedures (the “how”), timeframe (the “when”), the participants (the “whom”) and the rules (the “commitment”, including operative principles, financial, infrastructural, and human resources). The most important elements of the policy framework for the international network of ex situ base collections address: a. the strategy; b. the global plan of action; c. monitoring and reporting; d. the international network of ex situ base collections as the implementing agencies; e. the International Treaty and its MLS as an oversight body; f. the CGRFA that provides oversight, in close consultation with the treaty.

Components of the global system listed below contribute in their respective capacity to such a strategy, including the listed “political” elements that underpin the actions.

#### 5.1.1. The Global Strategy for Plant Conservation (GSPC)

The GSPC was adopted by the Convention on Biological Diversity (CBD) in 2002. It provides a framework for the policies and actions required to prevent the loss of plant diversity and promote plant conservation. Target 9 of the GSPC is closely linked to Aichi Target 13 [182], which addresses the key objective relating to genetic diversity and provides a clear entry point to the work carried out in the framework of FAO, its Global Plan of Action for Plant Genetic Resources for Food and Agriculture (see below) and the work of the CBD. While crop diversity is well represented in crop gene banks, crop wild relatives (CWRs) and other socio-economically important species are significantly underrepresented. In this respect, botanic gardens and other plant conservation organizations play an important role. The FAO Commission reports to the CBD on progress achieved with the implementation of the GSPC and participates in the post-2020 strategy. The GSPC addresses PGRFA through the implementation of the strategic plan of the FAO Commission. As such, it is at a rather high level for actual gene bank operations and impacts mainly at setting priorities for actions through the GPA, as mentioned below. The FAO agreed on a strategic plan for the Commission on Genetic Resources for Food and Agriculture (2019–2027), providing high-level strategic guidance to the commission and its member states [183]. There are no specific points to mention that directly impact our assessment as this strategic plan provides merely a framework of its components.

#### 5.1.2. Global Crop Conservation Strategies

For 26 food crops, global crop conservation strategies have been developed over the past 20 years. These global crop conservation strategies provide a very useful framework for prioritizing and planning collecting activities for the most important food crops, based on existing collections, on gaps in terms of geographical areas that are underrepresented in collections, for conservation approaches followed, including research and what needs to be performed based on latest knowledge and information to help plan and prioritize actions to ensure the long-term conservation and availability of plant genetic resources for food and agriculture. The evolution toward a rational global system necessitates the identification of the location and status of unique genetic diversity and the outlining of the processes by which this diversity can be most efficiently and effectively conserved and used for the benefit of the global community [184]. The strategies cover not only the crop species but also the related crop wild relatives and treat the different types of germplasm to some extent and make mention of in situ and on-farm conservation activities. They do set priorities for recommended actions and thus, are a particularly useful and important starting point for gene banks to consult when planning new activities that would include one or more of the food crops covered by these strategies.

The most common constraint in developing such crop strategies is the lack of sufficient accession- and collection-level information to categorize collections by importance, to identify duplicates, and to fulfill other tasks needed for the ideal strategy. In addition, the use of different taxonomic systems for the same crop and misidentification of accessions are reported problems to generate well-informed crop strategies [184]. Another constraint is obviously that only a limited number of crops are covered, that only information has been covered that was readily available to the crop experts, and thus, that the holdings of many (smaller) gene banks in several countries are not covered. At the same time, the strategies do provide ample information on gene banks holding major collections, existing networks, and other useful aspects. The global crop strategies represent a major undertaking in the field of plant genetic resources conservation, mobilizing experts to collaboratively design plans for more efficient and effective conservation and use of crop diversity ex situ. Supporting the development and updating of crop strategies allows moving forward toward a more efficient and useful global system of (long-term) conservation and use of these invaluable plant genetic resources [184]. The Crop Trust provides access to the existing 26 strategies and is in the process of revising and updating several of the existing strategies and developing new ones for 10 more crops/crop groups [185].

### 5.2. Global Plan of Action (GPA) for the Conservation and Sustainable Use of PGRFA (GPAII)

The GPA is one of the important instruments for countries and gene banks to obtain relevant and useful information about general conservation and use aspects, identified global priorities, reported needs and opportunities, etc., for 18 global priority activities. The GPA is the most important reference document for national, regional, and global efforts to conserve and use PGRFA sustainably and to share the benefits that derive from their use in an equitable and fair way. The strength of the GPA is that it squarely addresses all PGRFA, in a general sense, in situ and ex situ conservation, sustainable use, and building institutional and human capacity. It also provides guidance in linking conservation and use in achieving more sustainable production through breeding and broadening crop diversity. Strengthening of national programs, the promotion, and strengthening of PGRFA networks, constructing comprehensive information systems, and strengthening public awareness of the importance of PGRFA are important topics addressed. Furthermore, it was a very consultative process that had led to the formulation and adoption of the plan, involving experts and specialists for the main activity areas. The updating of GPAI was based on the first state of the world’s PGRFA (SOW) report [186]. The GPA also anticipates developments and trends in agriculture that might impact the conservation and use of PGRFA, e.g., the industrialization of agriculture, low input agriculture, the globalization of markets, including the seed sector, the increasing use of genetically modified varieties, the strategic use of PGRFA to better cope with climate change, major advances in science and technology, the advances in molecular and genomic methods as well as policy developments [187]. For more details, see also Section 2.5 of [3].

A possible weakness of SOWs to be guiding documents to gene banks and countries is that each (updated) GPA covers a relatively long period of 10 and more years, that it is a complex and rather bureaucratic process, and that its implementation is “voluntary”, with limited opportunities for monitoring its implementation. The lack of financial resources is one of the constraints to implement the GPA to its full extent.

The opportunities the GPA offers to governments, research, and conservation communities, as well as to the users of PGRFA, are directly related to the extent of its implementation. The more the GPA is used as a priority-setting mechanism, the better it will be implemented, and the more efficient and effective the conservation and sustainable use efforts become worldwide.

Reports on the State of the World’s Plant Genetic Resources for Food and Agriculture (SOWs) [186,188] present the outcome of periodic assessments by FAO of the state of the world’s PGRFA, thereby facilitating analyses of changing gaps and needs and, thus, contributing to the process of updating the “rolling” GPA. To facilitate this monitoring process, and thus the preparation of the SOW report, an online reporting tool of WIEWS has been made available [189] for the preparation of SOW III, scheduled for 2023. The monitoring for SOW II was conducted through a participatory process with the appointed national focal points in the member states that were guided to formulate a country report through a nationwide consultation and to use tools to guide this process. Ninety-one country reports have been prepared between 2014 and 2017 [190]. These country reports are particularly useful and important documents as they contain ample and typically detailed information on the state of PGRFA conservation and use in the respective country. The preparation process of the SOW report is possibly the most comprehensive global assessment on the state of PGRFA, in countries, in regions (in the past through regional consultation meetings), and globally and thus, the publications this process generates are indispensable resources for planning activities, for prioritizing efforts at the various levels and to obtain a better overview of the current situation of individual crops, crop gene pools as well as of countries. It should be noted that the individual country reports vary greatly in quality, but all follow a defined structure and can be (relatively) easily consulted. A major drawback might be that almost half of the countries did not produce a country report, and thus, the coverage of the country reports collectively might well be incomplete on aspects such as distribution of species, their conservation, and use status, etc.

### 5.3. International Treaty on Plant Genetic Resources for Food and Agriculture (ITPGRFA) and Its Multilateral System (MLS)

As the successor of the International Undertaking, the International Treaty with its MLS [191] is the most significant policy instrument that we have for PGRFA. For details on this legal agreement, operational since 2006, see the work of [3]. In November 2020, the treaty had 148 contracting parties, including the European Union. The treaty aims at establishing a global system to provide farmers, plant breeders, and scientists with access to plant genetic materials. Its main provision is the establishment and operation of the MLS, which puts 64 of the most important crops into an easily accessible global pool of genetic resources that are freely available to potential users in the treaty’s ratifying nations for specified uses. With the creation of the MLS and placing genetic resources of the above-mentioned 64 crops and species into the public domain, it was intended to make access to PGRFA easy and simple.

However, it should be mentioned that this access is somewhat “bureaucratic” as it is regulated by a relatively complex standard material transfer agreement (SMTA) that defines several conditions and restrictions, including for the use of such materials. Furthermore, despite the impressive number of contracting party states, many of them did not contribute materials to the MLS, thus limiting the total genetic diversity in the pool. Until mid-2019, a total of 58 notification letters designating PGRFA to the MLS were submitted to the secretariat of the treaty by 44 countries and six organizations. Furthermore, 18 international and regional centers have included their ex situ collections in the MLS [192]. In addition, the rather restricted list of 35 crops and 29 grass and forage species that have been included in Annex 1 of the treaty is a significant limiting factor. For instance, only a small number of vegetables and neglected and underutilized species has been included in the list, no commodity crops, etc. Consequently, for many countries, the current MLS is of limited interest. Yet, another limiting factor of the MLS is the hesitation of the private sector to use materials from the MLS, especially triggered by the benefit-sharing conditions. Because of the difficulty to ease the access conditions through negotiations at the Governing Body meetings, many of the bigger breeding companies have established their own private collections, partly officially through the MLS and partly through their own initiatives with gene banks and countries [193].

The strong link between the Global Crop Diversity Trust and the treaty is a major strength of the MLS. The Crop Trust has fully embraced supporting the major food crops and their wild relatives that are included in Annex 1 as their priority crops, but also realizes that other important and minor food crops would further strengthen the MLS and started a project to develop global crop strategies for gene pools that have not yet been covered so far. Since the Crop Trust is a major partner in the international network of ex situ base collections and as Genesys (initiated and operated by the Crop Trust) is one of the main information resources on global germplasm collections at the accession level, there is a direct link between effective and efficient long-term conservation of major food crops’ gene pools and the MLS.

### 5.4. The FAO Commission on Genetic Resources for Food and Agriculture (CGRFA)

The CGRFA provides a platform for all the PGRFA as well as for animal, aquatic, and forest genetic resources [194]. For details on the historical achievements of the Commission, see Engels and Ebert [3]. With the establishment of the International Treaty, replacing the International Undertaking, and the inclusion of other sectors of genetic resources, in particular forest, animal, and aquatic resources for food and agriculture, the role of the commission changed. However, as most of the food and agricultural crop species are not covered by the treaty, the commission continues to be of direct relevance as an oversight body for most of the PGRFA. The relevant provisions of the commission for the international network of base collections and global conservation system have been mentioned by Engels and Ebert [3], in particular, the Genebank Standards, the rolling GPA, WIEWS, and the SOW and country reports. In terms of global coverage of countries, in July 2014, 178 countries and the European Union were members of the commission [195].

The strength of the commission is that it has a broad scope and covers all genetic resources of relevance to food and agriculture, with a clear focus on food security and sustainability issues. Another strength is that most of the countries are members of the commission. A possible weakness is a fact that many member states do not place the conservation and sustainable use of PGRFA high on their agenda and that the commission as a forum for discussion has no possibilities for sanctioning members that do not implement agreed activities or do not adhere to agreed standards. Furthermore, the commission has to share the oversight over the international network of base collections with the treaty, and thus, a clear focus is somewhat lost. When looking at the global conservation system, it should also be noted that other elements, organizations, operations, and activities that directly or indirectly contribute to international collaboration on conservation and use of PGRFA are currently outside the global system, or their contributions are not fully appreciated, e.g., many NGOs, farmers’ organizations, botanic garden networks, and others. The global system does not provide many opportunities to strengthen linkages and connections between the existing components of the system. The global dimension of in situ conservation is not well developed. In addition, the financial arrangements that support the system are underdeveloped [196].

The following websites provide relevant information regarding the different elements of the international network of ex situ base collections:WIEWS—World Information and Early Warning System on Plant Genetic Resources for Food and Agriculture (http://www.fao.org/wiews/en/) [197];CGIAR Genebank Platform (https://www.genebanks.org/the-platform/) [198];Multilateral System of the International Treaty (http://www.fao.org/plant-treaty/areas-of-work/the-multilateral-system/overview/en/) [191];Svalbard Global Seed Vault (https://www.seedvault.no/) [199];Global Safety Backup Cryopreservation Facility, still being developed (http://www.fao.org/3/CA1371EN/ca1371en.pdf) [200]. The Global Safety Backup Cryopreservation Facility is an initiative that has been developed by many key players in the field of cryopreservation and coordinated by the CGIAR gene banks, as a parallel to the SGSV in Norway, to provide a global security backup for cryopreserved accessions. This mechanism is awaiting political acceptance;The European Cooperative Program on Genetic Resources (ECPGR; https://www.ecpgr.cgiar.org/about/overview) [201];GRIN-Global (https://www.nal.usda.gov/grin-global) [145] is based on the plant database of the Germplasm Resources Information Network (GRIN).

### 5.5. International Network of Ex Situ Base Collections

In the following, we will focus on the base collections maintained by the CGIAR centers, CATIE, and the Pacific community, as they have concluded agreements with the secretariat of the International Treaty and thus, formally included their designated germplasm in the MLS. In addition to these international centers, also the national collections in countries such as the USA, South Korea, Japan, Germany, The Netherlands, and some others are important partners in the MLS. The 5th largest international gene bank in the world, dedicated to the conservation of global and indigenous vegetable germplasm and maintained by the World Vegetable Center in Taiwan [69], is distributing germplasm globally under the SMTA, thus applying the guidelines of the International Treaty and the MLS but is yet to officially sign an agreement with the Treaty Secretariat.

The CGIAR collections, as well as those maintained by the World Vegetable Center and CATIE, are to a large extent collections with global coverage, have a strong focus on (major) crop gene pools, are relatively large, well-managed, and form the backbone of the MLS of the International Treaty. Furthermore, most of the centers that included germplasm in the MLS also have active breeding programs with a global focus and actively distribute breeding materials as well as newly bred varieties worldwide. The integration of the gene bank and its collections and the breeding cum research activities into a research institute has many advantages and has proven to work well. It should be noted that materials of other (minor) crops that are maintained by international and regional centers (Article 15 species) are also included in the MLS and thus available to all users through a standard material transfer agreement (SMTA).

The maintenance of global germplasm collections, usually with a suitable representation of the total genetic diversity, and targeted (mandate) crop breeding activities “under one roof” are important strengths that the international centers offer. Furthermore, the global coordinating role that these centers play with respect to the conservation of their respective mandate crops, in many cases following the elaborated activities in the respective global crop strategies, and the coordinated management of these collections with respect to gene bank standards, policies on access and distribution, information and research, through the Genebank Platform should be mentioned as another significant strength. In addition, the assurance of stable funding over the long-term, thanks to arrangements through the Endowment Fund of the Crop Trust, combined with the provision of training and advice to collaborating scientists from developing countries, this network of international centers is a true pillar of the international network [188].

The current international network of ex situ collections also has some obvious weaknesses. Possibly the most important one is the strong focus on the major food crops, i.e., those listed in Annex 1, and the relative neglect of minor (including global and regional) crops, such as vegetables and the locally adapted NUS or orphan crops. Another weakness is the, in general, less active involvement of smaller developing countries with collections consisting of mostly local minor crops, frequently enriched with breeding materials and varieties of the international research centers. As many of the centers of diversity are situated in developing countries, and since these are usually less well collected (possibly except for the mandate crops of the international research centers), there is likely also a bias toward the more accessible and bigger developing countries. Yet another factor that relates to the previous points is the lack of functional breeding programs, possibly also other research programs worldwide for (most of) these marginal crops, and thus, the countries that are “left out of the system” do hardly benefit from the genetic resources in their territories. However, more recently, some donors have recognized this aspect and initiated regional or continental programs or projects to address, for instance, the genetic diversity of African orphan crops [202] or to build molecular research capacity as part of an agricultural transformation program [203].

The Governing Body of the treaty is well aware of the limitations of the restricted coverage of Annex 1 crops and species causes, and for a number of years, negotiations have been ongoing to overcome this shortcoming. An “Ad hoc open-ended working group to enhance the functioning of the multilateral system” has been meeting nine times, but no results could yet be achieved [204].

For the CGIAR centers, optimal functioning of the MLS is critically important to fulfill their mission. Unfortunately, some challenges have been faced since its inception. Since 2018, only one payment has been made to the Plant Treaty’s benefit-sharing fund, in line with the mandatory monetary benefit-sharing conditions included in the SMTA that are triggered when breeding products derived from MLS germplasm materials are commercialized. Furthermore, many potential providers are demonstrating reluctance [16] to proactively provide access to plant genetic resources in the multilateral system until more money from commercial users is contributed to the benefit-sharing fund of the treaty. In addition, the treaty’s relatively low profile in many countries makes it difficult to obtain political support to implement national measures. Moreover, some companies and universities have declined to take germplasm materials from the MLS because of difficulties with the SMTA. Finally, very few requests for CGIAR germplasm come from farmers or farmers’ organizations, civil society organizations, or countries with small or no plant breeding programs for direct use [16]. In this context, it should be noted that direct use of materials from the MLS is not explicitly foreseen by the SMTA. In addition, other aspects that limit the transfer of germplasm from the MLS to users have been noted, including the uncertainties whether SMTAs are used when passing germplasm on to third party users, the lack of reporting evaluation results by recipients of the germplasm to the CGIAR centers, despite reminders. The same is the case with respect to feedback regarding improved lines that are being released by countries as cultivars and thus, no, or only limited value adding on the germplasm in the MLS takes place through information sharing [16].

Whereas through the establishment of the MLS, a global system is, in principle, available for placing and keeping (including for the long-term) PGRFA important for food security, nutrition, and sustainable production practices in the public domain, thus making those freely available, there is a need to further strengthen the system. One important issue is the restrictive and very limited list of species included in Annex 1; it should rather be all PGRFA. Furthermore, the benefit-sharing arrangements are too bureaucratic and complex, and they should be replaced by a system that altruistically supports the financially poorer but genetic diversity-rich countries with their conservation efforts. A global system could play a key role in this, especially through the establishment of close linkages between conservation and use. The international network of ex situ base collections could be the central conservation component of such a system, ensuring the effective and efficient long-term conservation of unique, carefully selected, and representative samples that are being made readily available to users worldwide. The efficient management of the related information is another key aspect of such a rational global PGRFA system.

### 5.6. Observations on Strengths and Weaknesses of Routine Genebank Operations, and Opportunities to Facilitate Cooperation between Genebanks at Large to Contribute to the Global System

In this subsection, we will assess strengths and weaknesses for each of the key routine gene bank operations vis-à-vis the relevant aspects of the global conservation system and, where opportune, identify and describe possibilities for more active participation of individual gene banks in the global system. In order to provide a more concrete context, global data are reported, sometimes repeating earlier facts that might help to understand this context better; they stem largely from the comprehensive but somewhat outdated SOWII report [188], from the Genebank Platform [198], and have been updated with personal observations and experiences, whenever possible.

#### 5.6.1. Exploration and Collecting; Prioritization

Whereas for most of the major food crops, a large part of the genetic diversity is represented in collections, for many other crops, especially many neglected and underutilized species, and CWR, comprehensive collections still do not exist, and considerable gaps (genetically as well as geographically) remain to be filled [188];Crop conservation strategies are possibly the most powerful tools to set priorities where to collect what as they typically provide a global account of existing collections, their diversity coverage (as far as known), reported threat status, etc. For details, see Section 5.1.2. For understandable reasons, there has been a strong focus on major food crops, in particular orthodox seed-producing crops. However, of the 10 new crop gene pools, several vegetables, and neglected and underutilized species have been included;About 55% of accessions that are being conserved in gene banks globally, and for which the country of origin is known, has originated in the country where the collection is maintained [188]. The lack of information on the other 45% of accessions is a serious limiting factor in planning as well as developing convincing crop strategies to guide collecting priority setting;Since the mid-1980s, the average number of accessions collected annually decreased [188]. The number of newly acquired materials by the centers to be included in the international collections dropped down to the lower levels that characterized the mid-1990s to 2009 [47];Largely triggered by particularly the legal access and benefit-sharing issues, a significant shift from internationally organized toward national collecting missions has been observed since the mid-1990s. This has resulted in slower growth of new accessions that are globally being made available. Thus, increased technical and financial support to national and local programs will be indispensable to ensure that timely collecting will be undertaken;Priorities for collecting at the local and national level should be on local and threatened genetic resources, whenever possible, with support from national or international crop gene pool specialists to ensure the highest possible quality sampling and treatment of the genetic resources;Collaboration with partners from outside a given country can be critically important to obtain specific expertise (e.g., taxonomic knowledge of targeted taxa for collecting) in preparing and conducting this demanding and complex activity;Sharing collected materials with other gene banks can be important to ensure effective and efficient long-term conservation if specific expertise is required for the conservation, e.g., cryopreservation, need for specialized infrastructure, and possibly other aspects.

#### 5.6.2. Conservation, General

Of the total 7.4 million accessions, national government gene banks conserve about 6.6 million, 45% of which are held in only seven countries [188];Of the analyzed 6,998,760 accessions, 10% were CWR, 24% landraces, 11% breeding materials, 9% advanced cultivars and 46% others and/or no information available [188]. A more balanced overall representation of some germplasm types is desirable;Total number of reported accessions in approximately 1750 gene banks or germplasm collections is about 7.4 million; of these, approximately 2 million are unique [205], and about 4.6 million are Annex I crops [188];Percentages of duplication within and between collections have been estimated to be high; only about 25%–30% (or between 1.9 and 2.2 million) are distinct [188];Many crops and important collections remain inadequately safely duplicated; for vegetatively propagated species and species with recalcitrant seeds, the situation is worse [188];Total number of accessions notified to be included in the MLS through a notification letter is 532,545, and 1,256,680 have been reported on the treaty website [192]. The number of CGIAR accessions reported in Genesys having been included in the MLS as of May 2021 is 766,680 [192];With respect to the security of stored materials, there are two main areas of concern, i.e., the extent of safety duplication and regeneration backlogs [188];Whenever possible, locally occurring or cultivated and thus, likely adapted germplasm should be locally/nationally conserved, either in situ and/or on farms, and preferably a copy of the material be included in an international collection as well. It would be ideal if such locally adapted and ex situ conserved materials could be regenerated/multiplied under local conditions that would allow a high-quality performance with limited risks of losing genetic diversity due to constraining conditions in the field. Where possible and/or necessary, collaboration with national or regional, and ideally with international experts and gene banks, should be sought;Priority setting at the local and national level will logically be based on criteria that are best known at the national level, especially regarding the threat status of taxa occurring in nature or on-farm;Whereas local limitations for conservation can be manifold, reasons and needs for participation in global (and regional) conservation efforts might be important and justified in case-specific infrastructure is missing, specialized expertise is not available, or local capacity constraints for storage and/or management exist;Participation of countries, including their relevant local gene banks, in the global system through the respective national PGRFA program is strongly encouraged and recommended in order to add value to and increase the safety of the nationally conserved genetic resources. This can be achieved through international collaboration but also through more sustainable “bottom-up” approaches;As for exploration and collecting, global crop conservation strategies do provide an excellent tool for planning and participation in global (and regional) conservation activities.

#### 5.6.3. Conservation—Linking in Situ and Ex Situ

As already implied in a recent publication by Engels and Ebert [3], linking in situ (including on-farm) conservation with ex situ conservation might well be indispensable to combine the strengths of each of these approaches and to complement the efforts at the local level with those at the national and international levels. This would ensure that a maximum amount of genetic diversity is conserved in the most appropriate and effective way and that biological and cultural information is not lost inadvertently [188];As in situ and on-farm conservation is per definition an activity that can only take place “locally”, it is a given that the coordinating national PGRFA program should provide the link between such in situ conservation efforts and complementing national, regional, or global ex situ activities;The important aspect of enabling effective and efficient (long-term) conservation applies predominantly to the local situation, i.e., there where the genetic materials occur in nature or on-farm and where also ex situ conservation should be applied. However, such complementary effort could also be applicable to situations where materials are being conserved “only” ex situ (and possibly under very constraining conditions for the germplasm itself) but where, for instance, a permanent evolution/adaptation of this material to changing conditions would be important;There is one more reason to actively promote collaboration between in situ and ex situ conservation efforts, and this concerns the facilitation of germplasm use. While direct access to populations or landraces conserved in situ is difficult, after collecting such material and storing it in a gene bank, access to those accessions by users can be targeted and routinely provided;Whenever genetic diversity occurring locally is being collected for conservation and use elsewhere, adequate arrangements should be made to ensure that benefits deriving from the use of such resources will be shared, in whatever form, with the local communities;There could be strong justification to combine the two conservation approaches in case materials are predominantly conserved in nature or on-farm, and a “strategic representation” of the genetic diversity in the gene bank would be important to ensure easy and targeted access to materials;Direct and effective links between germplasm conserved ex situ using techniques such as in vitro storage or cryopreservation and the corresponding materials maintained on-farm (and/or in field gene banks) will be important in case genetic instability is critical and regular “refreshment” of the ex situ conserved materials is required;The establishment and operation of local “community gene banks” might well be considered to strengthen local conservation capacity and to facilitate collaboration with the national (and eventually international) level and thus, to empower/strengthen the ownership over the local resources by the local communities [206].

#### 5.6.4. Germplasm Management, including Processing

An important prerequisite for effective and efficient germplasm and gene bank management is the formulation of clear conservation and use objectives. Knowing why and what to conserve allows better planning and priority setting [14];Strong germplasm (and gene bank) management capacity is a prerequisite for efficient and effective long-term conservation. Many of the routine operations provide opportunities for increasing the longevity of conserved germplasm, of improving the quality of the management and thus of the materials, facilitating the use of the materials by adding value through characterization and evaluation, etc. Such improvements will make the gene banks more attractive to play a more prominent role in regional and global efforts, to access more easily additional funding through projects, etc., and thus, to become increasingly more attractive as a partner in collaborative research activities locally, nationally, and internationally;A well-coordinated national conservation effort [207] enables the country to be cost-efficient and effective with its national conservation activities, including in situ and ex situ, to make strategic decisions on assigning conservation responsibilities, strengthening the link between conservation and use, participating actively and effectively in regional and global initiatives, etc. [207];The CGIAR gene banks have a well-developed germplasm quality management system [208], basically following the Genebank Standards [5], the division of active and base collections with the most original sample concept [5,67] and other aspects such as risk management and user satisfaction [209] and thus, could well provide guidance to national and local management efforts.

#### 5.6.5. Storage—Seed; Field; In Vitro; Others

Of the reported more than 1750 gene banks worldwide, about 130 maintain more than 10,000 accessions each [188];Of the reported 3.6 million accessions (about half of the global total), maintained by 488 gene banks and germplasm collections, about 6% is conserved in field gene banks; almost 1% is conserved in in vitro collections, and 0.2% accessions are cryopreserved. Another 1.458 accessions are “conserved” as DNA [209]. For the CGIAR gene banks, most accessions are held and distributed as seed; just 23,862 (3.1%) are conserved as tissue in in vitro and 29,122 (3.8%) in field gene bank collections [16];About 96% of all conserved accessions in gene banks are maintained as seeds [188,198];In CGIAR gene banks, about 18,500 or 2.5% of the accessions are cryopreserved or safety duplicated as in vitro samples [198];Germplasm of crops listed in Annex 1 of the ITPGRFA is conserved in more than 1240 gene banks worldwide, and they add up to a total of about 4.6 million samples. Of these, about 51% is conserved in more than 800 gene banks of the Contracting Parties of the ITPGRFA, and 13% is stored in the collections of the CGIAR centers [188]. The fact that 96% of accessions are being kept as seeds in cold storage, it can be expected that the “global system” is being dominated by the conceptual thinking that relates to this type of genetic resources, possibly at the detriment of the non-seed conserved genetic resources or those with recalcitrant seeds;However, many of the gene banks do maintain germplasm falling in two or more categories and not seldom managing big numbers of different species, as for instance, botanic gardens do;A useful tool for planning and/or rationalizing gene bank operations and germplasm management practices is the published guide to effective management of germplasm collections [14];In cases that gene banks do not meet the aspired thresholds in conservation, such as the number of seeds per accession, seed viability, etc., it might be advantageous to seek advice and possibly assistance from experts, either in the national network or in the regional or certainly the global network. CGIAR centers are well geared toward providing such assistance, including capacity building;As human and infrastructural capacities for the more demanding conservation methodologies such as in vitro and cryopreservation are often missing in small or national gene banks, collaboration with other gene banks/countries that do possess such facilities will enable access to required knowledge and resources.

#### 5.6.6. Safety Duplication

The Crop Genebank Knowledge Base is a slightly outdated but still a very resourceful tool for most of the routine gene bank operations, including on the safety duplication for seed, clonal, and in vitro materials [210];It is estimated that more than one-third of the globally distinct accessions of 156 crop genera stored in gene banks as orthodox seeds are conserved in the Svalbard Global Seed Vault, with high coverage of Annex 1 crops and of those crops for which there is a CGIAR mandate. Cereals and food legumes together constitute 87% of the accessions in the Seed Vault [211];As per the CGIAR criterion for mitigation of risk of loss, seed accessions in long-term storage should be safely duplicated in two external locations, one of which is the SGSV; on this basis, 73% of the seed accessions have been adequately secured against risks of loss [16];The number of accessions stored at SGSV is 1,081,026 seed samples from 87 gene banks and 66 countries [199];The safety duplication of accessions is an essential step to increase the security of base collection materials as this germplasm is unique and forms part of a global inheritance. However, this should also apply to local and national germplasm collections to avoid possible losses;Safety duplication does not have to be in all instances under long-term storage conditions, in particular when germplasm is maintained in field gene bank collections. The duplication of the collection (or part thereof) at another location or another gene bank might be sufficient;Arrangements for safety duplication are best made with and through the respective national PGRFA program to ensure adequate coordination, conclude proper agreements and use the existing network of gene bank contacts.

#### 5.6.7. Germplasm Health

The Crop Genebank Knowledge Base provides very useful information on plant health testing [212];In 2018 and 2019, the germplasm health units (GHUs) of the CGIAR facilitated 3900 events of international germplasm transfers from gene banks and breeding programs, reaching >100 countries per year. In this process, GHUs tested 453,972 samples and eliminated 6% of those that were pest-affected [213];The CGIAR gene banks reported that during 2019, out of the 717,693 total accessions, 76,766 accessions were tested on their health status [213];Germplasm conserved as part of the international network of base collections, in fact, stored and distributed from the related active collections, should be free of pests and diseases when being distributed, in particular of quarantine pests and diseases [54];In addition, during regeneration/multiplication activities, regular inspection of the cultivated accessions in the field/greenhouse will be important to ensure the health status of materials that are conserved for long-term storage or maintained for distribution. For the latter, IBPGR created a series of technical guidelines for the safe movement of crops [214];Even for well-equipped and staffed gene banks, this specific routine operation on germplasm health continues to be a challenge, and collaboration with specialized institutions nationally or internationally might be required to handle germplasm in conformity with the rules and regulations of the International Plant Protection Convention (IPPC) and National Plant Protection Organizations (NPPOs) to avoid distributing harmful pests and diseases with germplasm samples;In the case of local conservation efforts, e.g., nature conservation, on-farm management, and community gene banks, proper links to the national PGRFA system that can provide germplasm health assistance will be important. Such contacts can also be useful to introduce, for instance, germplasm with resistance to locally occurring devastating diseases from elsewhere.

#### 5.6.8. Distribution and Exchange

The Crop Genebank Knowledge Base provides a section on germplasm distribution, including related biological as well as legal aspects [215];Relevant information on the transfer of germplasm of 18 crops/crop groups with respect to import and export requirements, guidelines for the detection and treatment of relevant pests and diseases, and best practices for seed and clonal germplasm materials can provide useful assistance [216];The total germplasm distribution remained steady over the period from 1996 to 2007 at about 100,000 accessions each year, and it peaked in 2004 [203];Over the last 10 years, the CGIAR gene banks have distributed more than 1.1 million PGRFA samples to recipients in 163 countries, or 23% of all PGRFA samples transferred following the rules of the MLS [47,217]. Over the first 10 years of operation under the MLS of the treaty, the CGIAR centers distributed almost 4 million samples of PGRFA with over 47,000 SMTAs. This represents 93% of the reported global distribution of germplasm under the multilateral system [16];The CGIAR breeding programs were the source of an additional 66% (approximately 3.3 million samples) of the PGRFA transferred through the MLS in addition to the above-mentioned 23%. The remaining 11% of materials exchanged were transferred by organizations and individuals outside the CGIAR [47];The CG gene banks reported 1238 external germplasm requests during 2019, and a total of 45,941 germplasm samples were distributed outside the CG, belonging to 38,099 accessions [218];Germplasm exchange and distribution have been important activities of gene banks and possibly the main source of acquisition of germplasm for many gene banks around the world (also recorded as donations). This has certainly been triggered by the fact that the first (oldest) gene banks were all outside the centers of crop diversity, and thus, germplasm collecting missions were typical to foreign countries, costly, and demanding. Consequently, interesting germplasm from the main crops was in many cases of interest to many other gene banks, and over time, such samples were globally distributed, resulting in a significant duplication, certainly from a global perspective;As described above, careful and detailed planning of new collecting missions, also with a clear understanding of what had already been collected in the past (also by other gene banks), is an important step to avoid unnecessary duplication. This also applies to the acquisition of germplasm from other gene banks. Consequently, this check should possibly be best performed at the national level;Since there are many steps involved in germplasm management that might jeopardize the genetic constitution or authenticity of the sampled populations or landraces, there is a growing recognition to a stronger focus on genetic diversity-related aspects of individual samples/accessions and to avoid steps that put this at risks;The above also requires better and comprehensive information to be provided along with the exchanged or distributed samples or accessions;As for some other routine gene bank operations, also the exchange and distribution of germplasm from a given country, especially when numerous gene banks and collections exist in the country, is best carried out in a coordinated manner by and through the national PGRFA programs, as they know the rules and procedures, know the collaborating institutions, etc.

#### 5.6.9. Regeneration and Multiplication

The Crop Genebank Knowledge Base provides a section on germplasm regeneration, why, and how it is being performed [219];Guidelines for the regeneration of 16 predominantly minor crop plants are an important tool and can be found at the Crop Genebank Knowledge Base [220];With respect to the security of stored materials at the global level, FAO identified two main areas of concern, i.e., the low extent of safety duplication and the large backlogs with respect to regeneration [188];Regeneration (and multiplication) is a complex, demanding, costly, and rather risky gene bank activity, certainly from a genetic diversity point of view. Consequently, significant backlogs have built up in many gene banks, with all the risks this might have of losing materials due to loss of viability. In particular, CWRs present many problems and challenges for their regeneration, as many are cross-pollinating, produce little seeds, and are highly shattering [184];One of the major constraints is that germplasm, in particular crop wild relatives, obtained from other gene banks outside the country might not be well-adapted (e.g., photoperiod and ecological requirements) to the growing conditions of the gene bank in question, and this can result in losses of genetic information or even of entire accessions and may need international collaboration [184];The CGIAR gene banks reported that during 2020 of the 721,574 accessions maintained, 11,414 were regenerated and 68,616 multiplied [221];Information reported and gathered on almost 900,000 accessions for the period 2012–2014 showed that 18% had been regenerated, whereas 38% needed regeneration. For about 40% of those that were due for regeneration, an adequate budget was reported not to be available [209];As regeneration requires solid knowledge of cultivating a given species, it might well have advantages to seek cooperation with institutions that possess such knowledge and experience. For instance, the Dutch gene bank CGN developed such collaboration with interested plant breeders in the country to regenerate accessions for the gene bank according to the gene bank standards. The breeder concerned was entitled to keep a subsample of the regenerated materials and expected to return the harvested produce along with any pertinent information to the gene bank. The World Vegetable Center followed the same procedure with Asian breeding companies [222];The Crop Trust offered countries assistance with the regeneration of germplasm accessions in the country, during the operation of a big global regeneration project, and expected materials of these accessions to be included in the collection of the respective center for long-term conservation with the intention that this material would be made available to users under an SMTA;Regeneration of germplasm materials provides several opportunities to complement gene bank functions such as gathering characterization data, increasing seed stock, and eliminating diseased plants in accessions [184].

#### 5.6.10. Characterization and Evaluation

The Crop Genebank Knowledge Base provides very useful and practical information on the characterization of germplasm materials [223];Characterization is regarded as a routine gene bank activity, typically using characteristics that are highly heritable. It is an essential activity that facilitates proper management of the materials and provides a solid information basis for its use. In contrast, evaluation is a step that requires specific knowledge of the conserved species, usually advanced facilities, and it needs proper experimental design to eliminate the generally high dependence of the evaluated traits on the environmental circumstances;Characterization produces valuable agronomic and breeding data and allows the identification of unwanted duplicates in the collection and thus, provides the basis for rationalizing the collection(s) by eliminating unnecessary duplicates [184];Collaboration of the gene bank with, for instance, plant breeders or specialized researchers is a common approach to get accessions evaluated. However, it seems advisable for the gene banks to make clear agreements with the plant breeders as their willingness to return evaluation results seems to be limited;The extent of characterization of collections held by CGIAR centers and World Vegetable Center (a total of 585,193 accessions were analyzed) is 77%, varying from 17% to 88%, depending on the crop [188]. Please note that these percentages vary greatly from one species or crop group to another;The average extent of characterization and evaluation of national collections in 40 reporting countries for the main crop groups is 64% of almost 320,000 accessions, 63% of the 410,000 cereal accessions were morphologically characterized, and 65% of the 48,000 vegetable accessions [188];When characterizing local germplasm materials, it is important to use characters and traits that are of interest to the local communities and, whenever possible, to use descriptors agreed among gene banks and to engage the local community in this activity.

#### 5.6.11. Documentation

Proper gene bank documentation and information management are essential pre-conditions for any gene bank to be effectively linked with any larger conservation system, to share germplasm and related information, and to be able to effectively and efficiently conserve PGRFA;Many attempts have been made since the 1970s to develop gene bank information management systems to ensure connectivity and the easy exchange of information between gene banks and users. The advances of the Internet, the recent development of management systems such as GRIN-Global [145] have greatly improved the myriad of approaches by individual gene banks or national PGRFA programs;Greater standardization of data and information management systems is needed [188] to facilitate their exchange, analysis, and thus their use in setting priorities, facilitating monitoring, and allowing the creation of a more effective and efficient global conservation system;Unfortunately, information management in many gene banks remains weak, and thus, this situation does not support effective and efficient participation in coordinated national or international conservation efforts;Of the accessions maintained in CGIAR gene banks, 87% have passport or characterization data accessible online [16]. This figure is much higher than the average percentage of passport and characterization data of gene banks at large;Well-organized and comprehensive information management in a gene bank is the basis for efficient and effective conservation; for orchestrating the active and base collections efficiently; to monitor the viability of stored accessions timely and efficiently (and thus, among others, avoiding unnecessary regeneration efforts); providing a sound and solid foundation for decision-making; enables targeted rationalization of collections and conservation practices; is a prerequisite for local, national, regional and global collaboration and cooperation; facilitates targeted use of conserved germplasm material, etc.;As for conservation activities, especially those that are integrated into a bigger “system” or network, the availability and curation of high quality and comprehensive data at all levels is an indispensable prerequisite for effective and efficient collaboration, as has been mentioned already in Section 4.3.

#### 5.6.12. Research

Whereas research can be regarded to steadily improve and strengthen gene bank operations, it is not an essential requirement per se to conserve germplasm efficiently and effectively. However, as already mentioned above, several routine conservation operations require at least some applied research activities to identify, for instance, the optimum SMC of seeds of less well-known species, to adjust conservation procedures to locally prevailing conditions, etc.;In general, a gene bank that fosters the culture of research, of interacting with researchers to understand local genetic resources better, to seek solutions of farmers to improve sustainable production, etc. will play a more important role in local and national PGRFA conservation and sustainable use and thus contribute to sustainable development, to better incomes for farmers and to protect the environment;Active researchers in gene banks are also more attractive to others that seek collaboration, increase the willingness of governments to support routine and new activities, etc.;Whereas applied research might not require collaboration, more advanced research activities certainly do, as the infrastructure might be lacking, the right expertise has not yet been built, or for other reasons. In addition, research on important contextual aspects of the germplasm in question might further add value to the germplasm. Examples could be on gathering specific traditional uses of collected landraces, agronomic information obtained from local farmers, and others.

#### 5.6.13. Collaboration and Networking

As conservation of PGRFA is still a relatively new science, as conservation operations are frequently localized, not widely recognized as critical, and in general rather complex by nature, collaboration offers solutions for problems that can hardly be resolved in isolation. Such need for collaboration applies at all levels and offers opportunities to learn new aspects, to share strengths and overcome weaknesses, etc. For aspects related to international collaboration, see also Section 6;Greater efforts are needed to build a truly rational global system of ex situ collections. This requires particularly strengthened regional and international trust and cooperation [188], and, as observed by the authors, this has significantly decreased in most of the regions over the past 10 years or so (among others triggered by the disengagement of regional and/or crop networks coordinated for several decennia by Bioversity International and others);To improve the management of collections and to facilitate increased use of the germplasm collections as part of a network, documentation, characterization, and evaluation need to be strengthened and harmonized, and the data need to be made more accessible [188];As plant genetic resources for food and agriculture are global for many of the cultivated crop species and, to a much lower extent for the related wild species, it is obvious that some sort of coordination from a global perspective is indispensable to achieve effective and efficient rational conservation of PGRFA and to facilitate their sustainable use. Thus, it is advantageous for all involved in conservation activities to seek collaboration with others, certainly at the national, depending on the scope and objectives of the conservation and use activities at the regional level, and whenever possible through the national program with the international efforts;Participation in regional or global PGRFA networks has proven to be advantageous to all involved from a contributing, receiving, and capacity-building perspective;Rationalization of collections is best performed at the regional/international level, requires comprehensive data and information about the local accessions/collections, and is an important step toward more cost-efficient and effective PGRFA activities;Active engagement of the lower in the next higher level in PGRFA activities through networking is a key prerequisite to enable and strengthen collaboration. One or more focal points for key areas might be one way to organize such participation effectively;Active and inclusive engagement of individual partners and participatory approaches seem to be pre-conditions to facilitate open, transparent, and motivating participation in conservation activities.

#### 5.6.14. Human Resources, Infrastructure, and Financial Resources

Many countries do not have the adequate human capacity, funds, or facilities to carry out the necessary work to the required standards. Many valuable collections are in jeopardy as their storage and management are suboptimal [188];In view of the afore-mentioned point, also, in this case, cooperation with other gene banks and institutions involved in the conservation and sustainable use of PGRFA locally or nationally is a possible solution to strengthen human capacity, to share facilities, etc.Many countries lack the resources needed to maintain adequate levels of the viability of the materials conserved [188], thus resulting in frequent regeneration cycles with the related difficulties of geneflow, loss of materials due to weather, and other conditions, at a high cost, and others. Thus, networking at the different levels is an important step in overcoming problems, constraints and in building capacity;Capacity building is at the core of the CGIAR centers’ work. CGIAR research programs support about 1000 students in their BS, MSc, and Ph.D. degrees annually [16];Training and education at schools, high school, and university in PGRFA conservation and use are important premises to build a sufficiently strong force of human capacity, as well as to create a broad awareness;A comprehensive overview of strengthening institutions and organizations and building capacity for the conservation and use of agricultural biodiversity was compiled in 2017 [224];In order to participate effectively in local, national, or international networking, a country needs a minimum level of conservation expertise;Additional resources for ex situ conservation need to be mobilized. Greater efforts are required to raise awareness among policymakers and the general public on the importance of PGRFA and the need to conserve it [188];In case farmers and other members of a community actively engage in conservation (and use) activities, it seems to be beneficial to place the local activities in a broader context and to explain the basic principles of evolution, genetic diversity, breeding and improvement, genetic erosion and of conservation activities in general.

Without a minimum of well-trained and knowledgeable human resources, effective and efficient conservation activities are not possible. Thus, reaching such a minimum “threshold” is an important step to achieve this. Seeking the best possible means and ways to achieve this would be to consult and possibly collaborate with the next higher level and certainly with the national-level PGRFA program.

#### 5.6.15. Linking Conservation and Use

Stronger links are needed between the managers of collections and those whose primary interest lies in using the resources, especially for plant breeding [188];An aspect of strengthening the link between conservation and use is the provision of adequate and comprehensive information on the germplasm materials, either conserved in situ, on-farm, and/or ex situ in genebanks;Typically, access to the materials is easier when conserved ex situ involving healthy and vigorous seeds or propagules and in adequate sample size for the user (albeit that this size is always small, such as research materials but hopefully sufficiently large to represent the genetic diversity the accession entails);Yet, at another level, users also expect to receive materials that can be easily used. In case the requesting user would like to have an adequate representation of the diversity present in the collection, the gene bank might want to create a core or mini-core collection to facilitate the use. Such subsets of a collection might also be “constructed” of accessions that possess a specific trait. An alternative approach for heterogeneous accessions that possess a defined trait at the accession level could be to split such accessions into either pure lines or in single-seed descents and to keep such “pure” lines separate, evaluate such materials for individual traits, and offer those with the wanted trait to users. This practice could work for self-pollinating species and be a real service to users;Constraints in using conserved germplasm in gene banks include accession-level data existence and accessibility, quality and status of the material, policy and legal obstacles, and awareness/education/ outreach [184];Priorities outlined in the crop strategies to enhance user relationships with collections include refinement of collections for breeders (use of marker-assisted selection technologies, creation of advanced core and mini-core collections, pre-breeding, further work on identifying diseases and resistance), and strengthening or creating new relationships with other users [184]. Further aspects have been mentioned in Section 3.4.

#### 5.6.16. Genebank Standards

Another important operational element of this international network of base collections is the Genebank Standards, a formally endorsed set of standards for routine gene bank operations for the conservation of orthodox seeds, non-orthodox seeds, and vegetatively propagated plants by the FAO Commission [5]. They do provide the foundation for worldwide collaboration between gene banks, as in fact is being demonstrated by the gene banks of the CGIAR using the CGIAR Quality Management System [208]) and the European gene banks using AQUAS, the Quality Management for AEGIS [225].

Whereas the use of standards has been widely accepted, it is very difficult to provide actual data on percentages of accessions that are conserved under “standardized” conditions. Only a handful of gene banks have implemented a gene bank-level quality management certification system (e.g., ISO 9001:2000), e.g., CGN (The Netherlands), IPK (Germany), CRI (Czech Republic), and the CGIAR centers CIP and CIMMYT, whereas the other CGIAR gene banks have a strict and active quality management system in place. Unfortunately, many of the gene bank collections are maintained without clear adherence to the Genebank Standards for a range of reasons. However, to effectively engage in a region of global conservation effort, following agreed standards is a prerequisite that cannot be avoided. The importance of quality management is demonstrated in the context of operating a regional virtual collection, i.e., AEGIS, in Europe [225]. Further details on the quality management of gene banks are provided by the Crop Genebank Knowledge Base [226]. 

#### 5.6.17. International Information Sources on Conserved PGRFA

A third important operational element of the global system is an information management system that includes relevant information on the materials included in the international network of base collections. An inventory that was started by IBPGR in the late 1970s eventually provided the foundation for WIEWS, a global information system on PGRFA facilitating information exchange. It consists of a global network of national focal points, a registry of more than 17,000 national, regional and international institutes and organizations dealing with the conservation and use of PGRFA. The WIEWS database is populated with information from direct contributions made to the 2020 data assemblage by countries on the implementation of the GPA and, as of July 2021, has registered over 5.7 million accessions from 420 genera and 54,306 species, conserved under medium- (30.4%) or long-term conditions (61.6%) in 831 gene banks from 114 countries and 19 international/regional centers [227].

CGIAR centers, and IRRI in particular, are contributing to the creation of the Global Information System for PGRFA (GLIS) [228] under the framework of the Plant Treaty. Work on the GLIS has focused on the development of digital object identifiers (DOIs) as permanent, unique identifiers for PGRFA accessions. Through the CGIAR Genebank Platform, the global version of the Germplasm Resource Information Network (GRIN-Global) and Genesys [145] have been enhanced to accommodate DOIs and link with the GLIS server. The CGIAR gene banks have already assigned DOIs to 73% of their accessions as of 1 April 2018 [16]. Other important international PGRFA information sources are EURISCO, the European Search Catalog for Plant Genetic Resources [229], linked to Genesys; the Svalbard Global Seed Vault’s Seed Portal [199], also linked to Genesys; the Kew Millennium Seed Bank List [230]; and the Global Biodiversity Information Facility (GBIF) [150].

## 6. Suggested Measures to Achieve a More Rational, Effective, and Efficient Conservation System

In the foregoing, we have addressed the weaknesses of the current long-term PGRFA conservation and facilitating use system. Here we identify measures that should be undertaken to overcome those weaknesses with the aim to obtain a more rational, effective, and efficient global system. In this process, we have been guided by the following statement: “The ideal future genebank will be part of a rational, efficient, and effective system in which genebanks work in close partnerships with each other, and in harmony with the scientific and the policy dimensions of research-for-development, ensuring that the benefits of innovation reach those who most need them” [42].

The weaknesses, limitations, or deficiencies of the current system include:Many crop gene pools are not adequately represented in gene banks, especially those of neglected and underutilized species as well as of crop wild relatives, and/or the genetic diversity of crops is inadequately represented in the gene bank collections. Given the ongoing and accelerating threat of genetic erosion, there is a need to systematically collect henceforth neglected genetic resources, following priorities yet to be established;In general, there is a lack of long-term storage and supporting facilities. Gene bank management practices are weak, and severe regeneration backlogs have been reported;Many gene banks have weak information management systems in place with concomitant poor coverage of basic information on the germplasm conserved;Many gene banks do not have or use advanced genotyping and phenotyping technologies and do not collaborate with other gene banks having such facilities;Many gene banks have only weak or no linkages with the breeding community and other germplasm users;There are many unwanted duplicates within and between collections that do not need to be included in a long-term base collection system;Over the past two or three decennia, a decrease in regional and global coordination, as well as cooperation between gene banks, has been observed;The international distribution of germplasm materials included in the MLS as well as of breeding materials is primarily performed by centers of the CGIAR and AIRCA who have the technical expertise and resources to conduct adequate germplasm health testing and to eliminate pathogens that fall under quarantine restrictions. Unfortunately, most national gene banks do not have the facilities and expertise to fulfill the increasingly demanding requirements for the safe distribution of germplasm;The current legal and policy framework restricts rather than facilitates collecting, conservation, and distribution/exchange of threatened plant agrobiodiversity;The oversight and management of the global conservation system are weak, and there is no strong, active participation of many countries in the system.

From the above list, it becomes clear that the current arrangements and mechanisms need significant improvement and expansion to achieve efficient, effective, and rational global PGRFA long-term conservation and facilitated germplasm use. While the current global system has many of the required components in place, one must conclude that it is not functioning well, possibly due to a high level of bureaucracy as well as mistrust between diversity-rich, developing countries, and advanced countries with a stronger economy. This can be deduced from the current lack of progress in negotiations on the expansion of the list of Annex 1 species and the ongoing debate on how to solve the issue of digital sequence information sharing (see Section 4.4). More visible and tangible benefits, either monetary or otherwise, should become available to biodiversity-rich countries and communities that are willing to participate in such a system for the sharing of genetic resources and traditional knowledge.

The following measures might help to overcome some of the identified weaknesses and contribute to a more rational, effective, and efficient long-term conservation system with use facilitation.

Filling genetic and geographic gaps in current collections. While genetic diversity of major food crops is generally well represented in collections, underutilized crops and wild food plants, as well as CWR, are clearly underrepresented, and considerable gaps (both genetically and geographically) do exist. Many of the underutilized crops, wild food plants, as well as CWR, still play a critical role in achieving food security of the people in developing countries, especially in rural areas. Existing crop conservation strategies are powerful tools to set priorities where and what to collect as they typically provide a global account of existing collections, their diversity coverage (as far as known), reported threat status, etc. (for details, see Section 5.1.2). However, those crop conservation strategies are currently only existing for major crop gene pools and for only a few underutilized food crops.Determining unique accessions of germplasm collections at the global level. The total number of PGRFA accessions conserved in approximately 1750 gene banks or germplasm collections is about 7.4 million [188], yet only about two million of those are estimated to be unique [205]. With the current advances in genomics and phenomics, we envision genetic curation across international, regional, and national gene banks around the world to identify unique accessions across all crop collections or crop gene pools (see also Singh et al. [87]) and the European AEGIS approach [231]. Each unique accession would receive a globally unique ID, and duplicate accessions within and between collections could be removed, based on the decision of the holding gene bank, from long-term conservation in the respective base collections and could serve for germplasm exchange and user-oriented research. With such global curation, ex situ long-term conservation of crop collections and their wild relatives will become more effective and efficient.International collaborations. Intensifying existing international collaborations among institutes maintaining agricultural crop collections and forging new collaborations with the botanic gardens’ community offer important opportunities for a more rational, effective, and efficient long-term conservation system of plant genetic resources, their sustainable use, including the exploration of useful traits to strengthen breeding programs around the globe, thus, enhancing food and nutrition security of a growing population under the challenges of climate change. Intensification of collaboration with the private plant breeding sector would be yet another step to achieve this.Based on several case studies, Pearce et al. [232] highlight a number of key benefits of such collaborations, such as (i) synergy by bringing together collaborative teams with complementary skills and expertise; (ii) cost efficiency by sharing technologies and gaining access to local knowledge and other local resources; (iii) enhanced confidence for sharing and aggregating of resources such es accessions and specimens and associated information kept in globally dispersed collections; (iv) long-term positive change that brings about impact and leverage beyond the specific objectives of the collaboration; and (v) transfer of knowledge and technologies lead to building local expertise and strengthens national research capacity for plant diversity studies and conservation efforts. Furthermore, setting and applying standards, potentially extending plant breeding to “orphan crops”, and facilitating better coordination of activities would be other significant benefits that enable the emergence of a more rational and effective global system.Investment in research. Conserving and delivering germplasm resources “in the right way” requires investment in conservation research and user-oriented supportive research and optimization of routine processes, which may include automation. Research investment should be directed toward seed longevity studies and viability testing intervals of orthodox, intermediate, and recalcitrant seed species and of crop wild relatives as well as cryopreservation, as both areas are of critical importance to improving efficiency and effectiveness of ex situ conservation protocols and gene bank management [42]. Other examples of (applied) research activities are listed in Section 5.6.12.With a focus on facilitating germplasm use, targeted investments in characterization (highly heritable descriptors that are also used by breeders), evaluation (including “smart” phenotyping and genotyping) will be required, and collaborations among gene banks need to be forged as investments in corresponding equipment and expertise are significant and should be shared for effective use. To enable gene discovery studies, it is essential to phenotype the pure lines (derived from single-seed descent) that have been used for GBS. The resulting data needs to be professionally managed and packaged for easy use by breeders and other researchers [42].Digital quality information on accessions. A better accession-level description of the genetic diversity of crop collections maintained in gene banks (including genomic, phenomic, and ecological data) and easy access for users to quality data on conserved germplasm and associated metadata is critical to meet evolving challenges in crop diversity conservation and to facilitate more efficient use of those resources in breeding and research [42,76,233,234].Access to detailed digital information associated with each accession is increasingly becoming as important for breeders and researchers as to the physical material itself, although this is currently proving to be a very contentious issue [53]; see also respective references in Section 4.4).According to Sackville Hamilton [42], forward-looking gene banks should envision “a digital catalogue of the functional genetic variants existing in each accession, linked to the corresponding information on genomes outside the gene bank. This will enable DNA-based decisions on conservation and, in conjunction with knowledge of gene function and associated phenotypic data, on use”.Legal and policy framework. The current legal and policy framework has evolved “spontaneously” along with the global conservation system. Since the entrance into force of the CBD, and the recognition of national sovereignty with respect to biodiversity, its conservation, and sustainable use, a strong focus was given to issues related to a property right, as well as access and benefit-sharing. With the establishment of the International Treaty, the MLS was intended to play a central role with respect to ABS arrangements of germplasm that had been placed into the global system, restricted to materials under governmental control and in the public domain, thus largely confined to the species listed in Annex 1. From a PGRFA perspective, it would make more sense to aim at the coverage of all PGRFA and to avoid any exceptions.Whereas many countries are members of the FAO Commission on PGRFA as well as the International Treaty, the active participation of many countries in the debates in the Commission and Governing Body is limited, as is the level of implementation of the agreed actions. Furthermore, the lack of trust (possibly caused by Western dominance), impressive bureaucracy, the lack of clear assignments of responsibilities, and the lack of shared benefits deriving from the use of the genetic resources provided by the diversity-rich countries are some of the main impediments for a conducive legal and policy framework that need to be addressed to achieve an inclusive and effectively functioning global system.Oversight and management. The co-existence of the FAO Commission on PGRFA and the treaty as oversight and policy- and legislation-setting bodies are sometimes difficult to understand and lead easily to non-transparent operations. With the understandable focus on member states, many actively engaged entities in the management, conservation, and use of PGRFA feel not to be represented at “the table”, and this leads to distrust and tensions, both at the national as well as international level. Furthermore, the perceived non-committal role of the private sector makes it difficult for countries to freely share their resources and to arrive at an effective, inclusive, and rational oversight of any global PGRFA initiative and the global conservation and use system.

The following “model” for a functional global network or system of base and active collections is proposed:For each crop gene pool, or where meaningful a set of related gene pools, a virtual global base collection is created and coordinated by a carefully selected lead gene bank to ensure that the identified unique base collection accessions (whenever possible also the “most original samples (MOS)” that for any accession could be identified) are stored under optimal conditions for the long-term. Such a virtual global base collection facilitates cost-efficient conservation operations through strategic coordination; it requires the coordination of a global crop gene pool network, that among other responsibilities, develops and keeps the global crop conservation strategy up to date; the coordinating gene bank should have the facilities and personnel to manage the virtual base collection under agreed quality standards, including the management of the MOS accessions, where necessary through the support to the gene banks that physically hold base collection accessions; each gene bank conserving accessions of the global base collection will also manage the associated active collection samples/subsamples for regeneration and arrangements for safety duplication, and backup storage at the SGSV as well as for distribution; all activities will be conducted in close consultation and agreement with the lead coordination gene bank;It would make logical and practical sense that current CGIAR gene banks that do hold global base collections for their “mandate crops” would continue these “assignment(s)”, including the operation of their active collections;Regeneration of the base collection accessions will be carried out by the gene banks that physically curate the agreed “base collection accessions” when the viability has dropped below the agreed viability threshold or the stock below the minimum stock size; if cultivation conditions for to-be-regenerated accessions are better at one of the other gene banks of a given accession, they could be asked to regenerate;The gene bank holding base collection accessions take in principle responsibility for the regeneration of those accessions and use the samples of a given accession from the active collection for at most four regeneration cycles before turning to the MOS subsample from the corresponding base collection accession; the gene bank holding base and active collection accessions are also responsible for regenerating the MOS subsamples, in close consultation with the coordinating lead base collection gene bank;Characterization and evaluation activities are coordinated by the lead base collection gene bank and in principle implemented by those gene banks holding physical base collection accessions; obviously, characterization will be combined with the regeneration of the accessions, whenever possible; each gene bank holding base collection accessions will coordinate research activities on their accessions, whenever relevant with additional accessions, in close agreement with the coordination gene bank;The coordinating lead base collection gene bank, in close consultation with all the operational gene banks holding physical base collection accessions, looks after human resources training and capacity-building activities; the lead base collection gene bank represents the base collection for a given crop gene pool (or as a “general” responsibility, with other lead crop base collection gene banks) in the network, and triggers policy and legal framework related activities;The base and active collection gene banks will conduct the distribution of those accessions they hold physically, if applicable upon request from the coordinating lead base collection gene bank, as the latter manages the global conservation as well as the global user databases for that crop and is the proposed recipient of such requests from users. The base and active collection gene banks are responsible for the germplasm health aspects of those accessions they are physically managing;It is foreseen that the “services” that gene banks provide for the management of the assigned base collection accessions will be paid for through the coordinating lead base collection gene bank from a to-be-formed global base collection fund, similar to the arrangements the Global Crop Diversity Trust has made with the CGIAR and a few additional gene banks;Countries/gene banks that participate in the global crop conservation network, e.g., through the inclusion and management of unique accessions of the base collection, and/or agree to conserve materials of the crop gene pool in situ or on-farm, or provide other “services” to the network, are entitled to participate in the sharing of benefits (e.g., monetary, non-monetary, membership in global and regional networks) that are derived from the network activities;All germplasm materials that are included in the global base collection network are “automatically” part of the MLS of the International Treaty and thus, in the public domain and freely available to all users without restrictions, as set out in the yet to be adjusted standard material transfer agreement with the stakeholder community of the group of global base collection networks. The International Treaty will be recognized as the global policy setting mechanism, but for all PGRFA; andA lean international organization, e.g., the Governing Body of the International Treaty, should assume responsibility for the global coordination, facilitation, and oversight over the various global base collection networks.

## 7. Conclusions and Recommendations

The current global conservation system has inherent weaknesses and limitations, partially due to its spontaneous creation out of a felt need by concerned scientists and visionaries and the subsequently required adjustments to the evolving political framework and changing realities. Here we recommend some measures that might contribute to a more rational, effective, and efficient long-term ex situ conservation system. These measures include:Filling genetic and geographic gaps in current ex situ collections through continued collecting of threatened genetic diversity with the aim of reaching an adequate representation of crop gene pools in ex situ collections, especially of neglected, currently underutilized species and crop wild relatives;Determining unique accessions of germplasm collections in base collections at the global level and removing the many duplicates within and among gene banks from the base collections. Identified duplicates could be used for research and germplasm distribution, rather than resorting to the base collection accessions;Intensifying existing international collaborations among gene banks maintaining agricultural crop collections, and forging new collaborations with the botanic gardens’ community offer important opportunities as well as stronger linkages with the plant science research community at large;The private sector, in particular the plant breeding community, is highly dependent on genetic resources and genetic diversity, and thus, their involvement in the conservation and sustainable use activities is desirable. Furthermore, plant breeders have specific knowledge of “their” crops and have the required expertise for regenerating and evaluating gene bank materials; hence, it could serve as a key link between long-term conservation and sustainable use. At the same time, it is obvious that adequate benefit-sharing arrangements, in the broadest possible sense, will be required to allow and strengthen this collaboration;More investment in conservation research and user-oriented supportive research, as well as optimization of routine gene bank processes, will help with conserving and delivering germplasm resources of high quality and in the right form as required by the users;A better accession-level description of the genetic diversity of crop collections maintained in gene banks (including genomic, phenomic, and ecological data) and easy access for users to quality data on conserved germplasm and associated metadata is critical to meet evolving challenges in crop diversity conservation and to facilitate more efficient use of those resources in breeding and research;The legal and policy framework clearly requires improvements, ideally by including all PGRFA in the MLS of the International Treaty and reaching a more equitable sharing of benefits derived from germplasm use, benefiting those who most need them and most often are the custodians of the rich agrobiodiversity on which people around the globe rely to satisfy food and nutrition security;A model for a functional and efficient global network of base and active collections and a lean international organization is proposed that assumes responsibilities for the global coordination, facilitation, and oversight over the various global crop gene pool base collection networks. This model should build on the existing gene banks of the CGIAR, the World Vegetable Center, and ICBA, as well as on a handful of strong national gene banks that form the core of the current global system on plant genetic resources for food and agriculture (PGRFA);The political oversight over the proposed global model network of base collections should remain with FAO and the Governing Body of the International Treaty.

## Figures and Tables

**Figure 1 plants-10-01904-f001:**
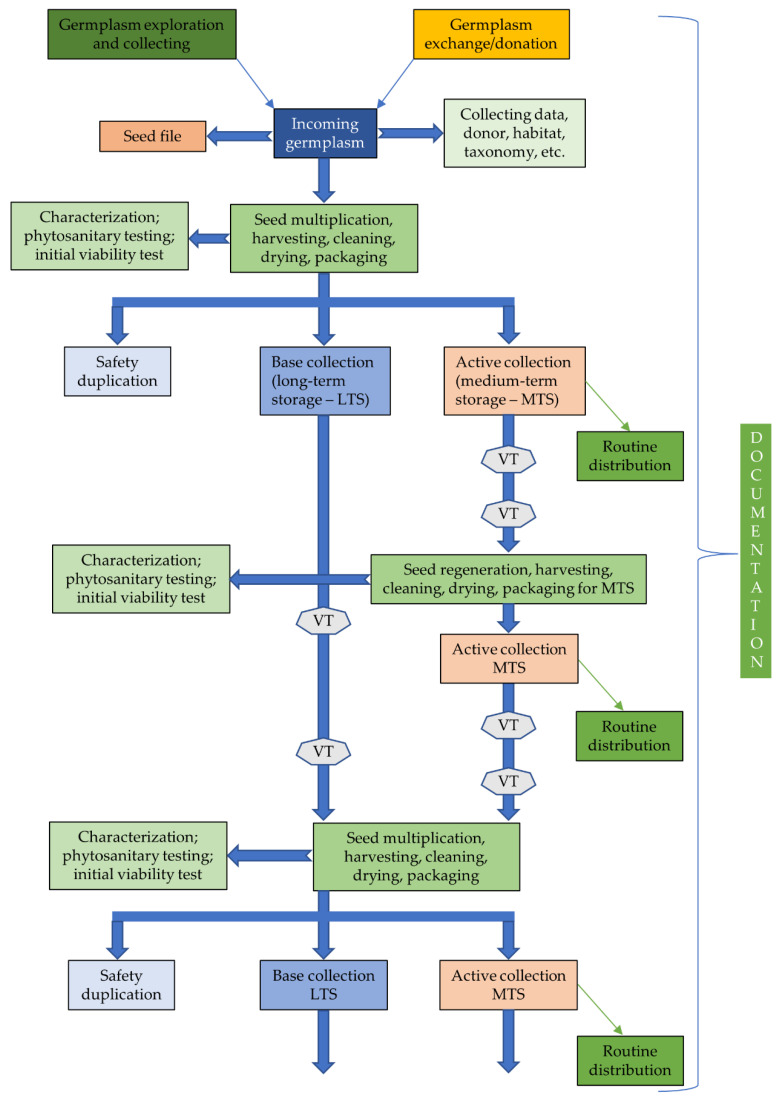
Overview of standard operations in a seed gene bank, based on the FAO Genebank Standards [5] and adapted from Hay and Sershen [6]. Germplasm arriving at a gene bank through a collecting mission or exchange/donation from other collection owners is assessed for its uniqueness, documented with all available information (passport data), and if seed quantity allows a sample of 20–30 seeds is separated to create a seed file for future reference. As the seed amount received is usually insufficient for storage as a base and active collection, the incoming seed lot needs to undergo a seed multiplication phase, followed by careful seed processing. During the multiplication phase, plants and seeds are usually characterized according to crop descriptor lists based on heritable morpho-agronomic traits. Seed subsamples will be taken for phytosanitary testing and for an initial viability test. Each subsequent regeneration event offers the opportunity for additional/complementary characterization and requires phytosanitary testing and an initial viability test. After equilibrium drying, usually at 5–20 °C and 10–25% RH, seeds will be hermetically packed for long-term storage (base collection) at −18 ± 3 °C and medium-term storage at 5–10 °C (active collection). Safety duplicate samples will be sent to another gene bank for long-term conservation under a “black box” agreement and/or to the Svalbard Global Seed Vault. Due to the higher storage temperature under medium-term storage conditions, viability testing (VT) must be conducted more frequently compared to long-term storage conditions. Should the viability of the active collection fall below 85% of the initial viability or seed quantity become insufficient, a seed regeneration cycle must be programmed for replenishment/replacement of the active collection. Up to four regeneration cycles can be conducted with seeds from the active collection before seeds from the base collection should be used. At the time of replacement of the seeds in the active collection, the viability of the seeds in the base collection needs to be monitored independently as seed lots are now derived from different regeneration cycles. Once seed viability of the accessions in the base collection falls below the threshold value, a new regeneration cycle with seeds from the base collection is required, following the same steps as described for the first seed multiplication cycle. This includes a replacement of the safety duplicate samples as their viability might also have fallen below the critical threshold value. Across all gene bank operations, a huge amount of data is generated, which needs to be captured and adequately managed in the gene bank information system for in-house use as well as for the benefit of germplasm users (passport, characterization, and evaluation data).

**Table 1 plants-10-01904-t001:** Examples of how (routine) research activities can underpin optimum management of germplasm accessions and collections.

Research Activities	Description
Determine genetic diversity of an accession	In-depth characterization of accessions using internationally agreed descriptor lists, making use of advanced molecular, genomic, and phenotyping tools.
Optimal management of gene bank accessions	Determine adequate, minimum numbers of individuals per accession for viability testing, regeneration, characterization, evaluation, and other gene bank activities with the aim of preserving the original genetic diversity.
Elucidate flower biology of crop species, when not known	Full understanding of the flower biology of a given species helps to avoid cross-pollination during regeneration and to maximize high-quality seed production for subsequent long-term storage.
Optimize seed production procedures	Genebanks conserving a wide range of different species (e.g., The World Vegetable Center) may need to conduct research to gain insight in crop-specific knowledge for optimizing seed production procedures to improve initial seed quality and, consequently, seed longevity.
Optimize seed drying procedures	The FAO Genebank Standards include clear and specific recommendations on all routine gene bank operations, including seed drying. However, rice accessions, for example, require modified drying procedures to enhance seed longevity (see Section 2.2).
Determine optimum seed moisture content (SMC)	Research has shown that SMC levels aiming at maximizing seed longevity differ among species. Thus, gene banks should consider conducting their own research to determine the species-specific optimum.
Optimize species- and accession-specific seed viability monitoring	Optimizing the schedule and procedures of routine viability monitoring of long-term stored accessions provides an early warning for deteriorating accessions and helps to rationalize the number of seeds used per test. Comparing the physiological response and storage behavior of seed lots produced in different crop seasons and/or environments improves our understanding of seed longevity. Weekly scoring of germination during a viability monitoring test helps to obtain information on seed vigor and how vigor declines as seeds age.
Predicting seed longevity	Conducting studies on the integrity of DNA and RNA in seeds under long-term storage helps to predict species- and accession-specific seed longevity.
Optimizing the genetic diversity representation of populations	Extending the knowledge of genetic parameters that allow optimizing the genetic diversity representation of populations of a given species will improve germplasm collecting and the establishment of truly representative collections.
Publishing research results	Publishing research results on the above-described topics in scientific journals would benefit staff from other gene banks and would boost the reputation of the gene bank itself.

**Table 2 plants-10-01904-t002:** Reasons for adjustments and refinements of the concept of active collections to optimize long-term conservation.

Topic	Description
Refrigeration issues	The cost of refrigeration, the unreliable electricity supply, and/or lack of adequate maintenance and repair opportunities of the cooling equipment may hamper adequate long-term conservation.
Overly large active collections	Many gene banks try to maintain overly large active collections (e.g., too many samples and subsamples of the same accession that stem from different regeneration cycles); the anticipated use of the materials is frequently over-estimated, and due to “sub-optimal” storage conditions for such accessions, avoidable higher regeneration frequencies are the consequence of keeping seed viability at the desired level.
Rationalizing collections	Improved germplasm management technologies, such as the use of barcodes, molecular tools, and digitalized information management, including early warning systems, can facilitate more effective and efficient gene bank management and allow to rationalize collections, e.g., sorting out genetic duplicates, removing accessions from the active collection that are never requested or used but are included in the base collection.
Accession management	Regenerated materials of a given accession are kept in the active collection under medium-term storage conditions. In case the regenerated subsamples do not suffice for further distribution or use, one could continue to use a regenerated subsample for a maximum of four regeneration cycles (and possibly consider regenerating more materials in case of high demands) before returning to the primary MOS from the base collection for the next regeneration cycle.
Optimizing seed management and storage procedures	Genebanks should carefully consider under which conditions to store the active collection. Lower seed moisture content and lower storage temperatures would result in much-prolonged storage periods with less total operational costs and increased (genetic) security due to reduced regeneration frequencies. Certainly, accessions that have low distribution numbers due to a lack of accession-level data could best be maintained under the same storage conditions as the base collection to avoid more frequent regeneration cycles triggered by a drop in seed viability. Maintaining those materials under long-term storage (base collection) conditions, materials would still be available for distribution, and costly regeneration cycles could be reduced [36].
New conservation technologies	Because of the rapid development of in vitro gene bank conservation techniques and the wider availability of cryopreservation protocols [68], germplasm materials previously conserved under more threatening conditions in field gene banks can now also be maintained as tissue in in vitro collections, thus increasing the security of the material, and/or be cryopreserved, and thus adding a long-term cryopreservation perspective.
Complementary conservation approaches	The increased availability and use of complementary conservation approaches and methods increase the overall security of accessions and allow to opt for the most effective combination of methods, both from a management as well as an economic perspective.

**Table 3 plants-10-01904-t003:** Weaknesses of existing base collections and options to overcome those weaknesses to arrive at a more secure and rational long-term conservation system of PGRFA.

Topic	Description
Poor representation of genetic diversity of natural populations and landraces in gene bank accessions	Original collecting often resulted in accessions that represented only a fraction of the prevailing genetic diversity of a population or landrace. In case sampling was performed of natural populations, one could consider lumping samples/accessions of the same population into one. In the case of landraces, one could consider lumping the samples collected from the same field.
Accession duplicates	In case one or more genetic duplicates of a given accession are identified in the collection, duplicates could be lumped into one accession. However, if an identified duplicate accession is phenotypical of special interest and has substantial research/evaluation data, it might be justified to keep that one separate.
Rationalizing base collections	When two or more subsamples of the same accession are found in the base collection (possibly from different regeneration years), the gene bank may opt to identify the most original sample (MOS) among them and to proceed with that subsample while moving the remaining subsamples to the active or the archive collection (see Section 3.3).
Reducing seed viability testing	A practice that is being recommended at the Center for Genetic Resources, the Netherlands (CGN) to rationalize routine gene bank operations is the decision to delay the first germination monitoring tests to 25 years after regeneration [72] or to the time when the samples of the active collection are undergoing the first regeneration cycle, and the origin of seed lots start to diverge between active and base collection (see Figure 1).
Base collection concept for vegetatively propagated materials	Whereas we have focused in the above completely on orthodox seed-producing species, it is understood that the concept of base collections might not apply to vegetatively propagated materials directly, commonly maintained in field gene banks due to the lack of available in vitro and cryopreservation options. In cases where in vitro techniques and cryopreservation protocols are available, the concept of base collections might apply as well. In all other cases, suitable maintenance in the field and adequate safety duplication might be the only option.

## Data Availability

Not applicable.

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
