# Peer review of "A Critical Review of the Current Global Ex Situ Conservation System for Plant Agrobiodiversity. II. Strengths and Weaknesses of the Current System and Recommendations for Its Improvement"

_plants, 2021, doi:10.3390/plants10091904_

Round 1
Reviewer 1 Report
In this manuscript, the authors review the current global ex situ conservation system for plant agrobiodiversity. It is a big review paper with several topics of ex situ conservation system. This manuscript looks like a guideline for plant agrobiodiversity. For me, it looks really important for food even related to SDGs. However, this manuscript is too long and divergence. I have no idea how many words are in this manuscript, but I think it is almost 30000 words. It means the authors need to concentrate their manuscript and make some tables, flowcharts, and figures. It will make it readers easier to understand their points.
Author Response
Thank you (Reviewer 1) for your positive comments. We appreciate your assessment and hope that the paper will contribute to food security and the implementation of the SDGs. Regarding its length, we have tried to delete less important paragraphs, to rearrange the order of some sections (i.e. documentation) and to guide the readers with help of a flowchart that addresses the important routine genebank operations. Furthermore, we have included three tables to replace long lists of bullet points.
Reviewer 2 Report
This very thorough and extensive review is well organized, well documented, and very very long. I think it is a good contribution to the field.
I think a bit more discussion about the predicted longevity of such collections would be important.
Likewise, this paper seems to strongly emphasize seed bank collections, although there is some limited discussion of those resources that cannot be seedbanked. But -- I do not wish to lengthen an already very long paper.
I believe this paper can be published.
Author Response
Thank you (Reviewer 2) for your very positive and nice general comment on our paper. We appreciate this and have full understanding for your comment regarding its length.
We have tried to delete some paragraphs that were possibly superfluous or partly repetitions of points already made earlier.
The discussion about seed longevity and its prediction has been expanded in section 2.3 and a reference has been made to section 4.1 where molecular approaches to dissecting seed longevity are discussed. Furthermore, wherever applicable, we have strengthened points with respect to seed longevity.
Reviewer 3 Report
The manuscript is a continuation of the previous paper entitled: “A Critical Review of the Current Global Ex Situ Conservation System for Plant Agrobiodiversity. I. History of the Development of the Global System in the Context of the Political/Legal Framework and Its Major Conservation Components” (published in Plants, https://www.mdpi.com/journal/plants/special_issues/Plant_Genetic_Resources_Conservation)
The review is interesting due to the complex approach and the multitude of related issues. Since the assumed objective intention of this manuscript was “to describe the role of active and base collections and the importance of linking germplasm conservation and use, also in view of new developments in genomics and phenomics that facilitate more effective and efficient conservation and use of plant agrobiodiversity”, there are many and varied approaches without full ‘coverage’. Obviously, due to the complex subject and space considerations, they could not be treated in more detail. However, the framework of the manuscript is in general properly and consistent in content, and the examples and the literature resources are relevant.
Please find below some general remarks or suggestions for possible completion or increase of the relevance and understanding of the manuscript by the readers:
- The manuscript is very extensive and it does not contain figures or tables that summarize some aspects presented (probably some of the text could have been compressed into appropriate tables, see: “examples on how (routine) research activities can underpin optimum management of germplasm accessions and collections, and thus, e.g. contribute to extending seed longevity of accessions”, Lines L505-540; “several specific reasons” updated and expanded by Sackville Hamilton et al. [65], that would justify a more critical assessment of the traditional concept of ‘Active Collection’, L620-648; “a close and critical look at the functioning of base collections for the secured and rational long-term conservation of PGRFA, assess possible weaknesses and suggest ways to overcome these”, L723-766 etc. See also L2265-2278, and correlate the institutions to the text above.
However, even though it is very interesting in content, the manuscript is very dense and may be difficult for readers to follow, or not very attractive to them.
- Therefore, it would probably be useful to have at least a figure, as schematic presentation of the conceptual framework of the whole manuscript, in which to present the approached elements and the interaction between them.
Or, if this is difficult to achieve one figure for the content, at least a few schematic figures on independent topics (or per chapter/section) could be used. For example, for chapter two, please see its contents below according to your manuscript, a simple scheme would improve the visual content of the presentation and assimilation by the readers of the information and its logical frame.
- Brief Description and Critical Review of Key Routine Germplasm Conservation Activities
2.1. Exploration and Collecting
2.2. Processing
2.3. Seed Longevity
2.4. Documentation
2.5. Seed Regeneration
2.6. Germplasm Exchange
2.7. Research
Speaking of the structure of the manuscript, and the chapters: here (see content of the section 2), probably the ‘Documentation’ sub-section would have been suitable at the beginning, before the first approaches (“Exploration and Collecting”). Please review the entire manuscript to ensure a clear structure, including the flow and logical course of the processes described (see also “5.6.11. Documentation”. Should not this sub-section be at the beginning of that section?).
Other aspects
Maintaining germplasm banks and avoiding the risks of biodiversity loss are discussed accordingly. Maybe other examples or approaches would probably be relevant, topical, and welcome (mentioned and discussed very briefly):
A) The risks due to public decrease in funding and personnel plant conservation and breeding programs (see Coe, M.T., Evans, K.M., Gasic, K., Main, D. 2020, Plant breeding capacity in U.S. public institutions. Crop Science, https://doi.org/10.1002/csc2.20227).
B) The risks due to environmental, climatic, or natural disaster factors, etc. difficult or impossible to control. Check: Svalbard Global Seed Vault - flooded after permafrost melts (the example of 2016, when the average temperatures over 7 °C above normal were registered on Spitsbergen, so the permafrost got over the melting point) etc.
There are discussions about ‘botanic gardens’ since the ‘Abstract’. Useful and recent sources for this topic and their importance in the manuscript context, as well as for ‘orthodox seeds’, ‘Millennium Seed Bank’ Kew etc., could be:
Liu, U., Cossu, T.A., Davies, R.M., Forest, F., Dickie, J.B., & Breman, E. (2020). Conserving orthodox seeds of globally threatened plants ex situ in the Millennium Seed Bank, Royal Botanic Gardens, Kew, UK: the status of seed collections. Biodiversity and Conservation 29:2901-2949. https://doi.org/10.1007/s10531-020-02005-6
Kovács, Z., CsergÅ‘, A. M., Csontos, P., & Höhn, M. (2021). Ex situ conservation in botanical gardens – challenges and scientific potential preserving plant biodiversity. Notulae Botanicae Horti Agrobotanici Cluj-Napoca 49(2), 12334, https://doi.org/10.15835/nbha49212334
L1597-1598:
“The following websites provide relevant information regarding the different elements of the international network of ex situ base collections: …”. It is preferable to facilitate the reader’s task and to include the links directly after the presented institutions (L 1599-1612) in the text. Obviously, references can remain to provide a complete description of the citations.
Please write ‘in situ’ and ‘ex situ’ using italics.
Author Response
The review of Reviewer 3 has been appreciated very much by the authors as it provided more specific suggestions and comments, besides the kind and positive words on the general content of the manuscript.
To facilitate responses to the various points we have copied the most essential statements of the reviewer into bullet points, followed by our responses.
- The manuscript is very extensive and it does not contain figures or tables that summarize some aspects presented (probably some of the text could have been compressed into appropriate tables:
Response: To provide a better guidance to the readers we have inserted a flowchart of the chronological sequence and the interactions of the routine operations of a genebank (i.e. Figure 1) and referred to it throughout the text. The three parts of the text for which the reviewer suggested to use tables instead of bullet points have been converted into tables.
-
... the manuscript is very dense and may be difficult for readers to follow, or not very attractive to them:
Response: Through the insertion of a flowchart and three tables, the text has become less dense and thus, might appeal more to readers.
-
Therefore, it would probably be useful to have at least a figure, as schematic presentation of the conceptual framework of the whole manuscript, in which to present the approached elements and the interaction between them:
Response: We appreciate this suggestion very much and have added a flowchart (i.e. Figure 1), containing the main routine operations of genebanks and their interactions, to provide an overview of genebank activities and thus, to guide readers through the text.
-
Speaking of the structure of the manuscript, and the chapters: here (see content of the section 2), probably the ‘Documentation’ sub-section would have been suitable at the beginning, before the first approaches (“Exploration and Collecting”). Please review the entire manuscript to ensure a clear structure, including the flow and logical course of the processes described (see also “5.6.11. Documentation”. Should not this sub-section be at the beginning of that section?):
Response: As mentioned above, we have attempted to follow the chronological sequence of a standard genebank as the basis for the presentation of our points in the paper.
All genebank operations generate data which need to be captured and documented to guide genebank management and inform users of germplasm. The documentation subs-section could be inserted at the beginning or as a final step in the routine genebank processes. In most cases, documentation is mentioned at a later stage (see also FAO Genebank Standards). Therefore, we decided to move the documentation sub-section (2.4) further down, now listed as sub-section 2.6, in agreement with the position of the second documentation sub-section under 5.6.11.
- A) The risks due to public decrease in funding and personnel plant conservation and breeding programs (see Coe, M.T., Evans, K.M., Gasic, K., Main, D. 2020, Plant breeding capacity in U.S. public institutions. Crop Science, https://doi.org/10.1002/csc2.20227).
Response: We do agree with the importance of reliable (public) funding and staffing of genebanks and, consequently strengthened the statements related to funding and staffing of genebanks in general. The excellent reference of Coe et al., 2021 has been added to the manuscript.
- B) The risks due to environmental, climatic, or natural disaster factors, etc. difficult or impossible to control. Check: Svalbard Global Seed Vault - flooded after permafrost melts (the example of 2016, when the average temperatures over 7 °C above normal were registered on Spitsbergen, so the permafrost got over the melting point) etc.
Response: The point regarding the increasing temperatures at Svalbard and the threat to the Seed Vault has been added to the section with a brief description of the Global Seed Vault.
- There are discussions about ‘botanic gardens’ since the ‘Abstract’. Useful and recent sources for this topic and their importance in the manuscript context, as well as for ‘orthodox seeds’, ‘Millennium Seed Bank’ Kew etc
Response: The authors fully agree with this point and are very much aware of the importance of botanic gardens for the ex situ conservation of PGRFA. Consequently, they have invited colleagues from the Millennium Seed Bank and BGCI to submit a paper on the approaches of botanic gardens with respect to PGRFA conservation to this Special Issue. Please also note that the reference of Liu et al., 2020 has already been included in part 1 of this paper as reference number 92.
- “The following websites provide relevant information regarding the different elements of the international network of ex situ base collections: …”. It is preferable to facilitate the reader’s task and to include the links directly after the presented institutions (L 1599-1612) in the text.
Response: The respective links have been included directly after each institution and the references have been maintained.
- Please write ‘in situ’ and ‘ex situ’ using italics.
Response: The use of italics for in situ and ex situ might vary with the journal in question. As the journal Plants is accepting the use of normal letters for these terms and as we used normal letters during our first article in this series, we plan not to use italics here.
Round 2
Reviewer 1 Report
The authors revised their manuscript according to my opinion and I don't have more comments.